# The Cas4-Cas1-Cas2 complex mediates precise prespacer processing during CRISPR adaptation

**Hayun Lee, Yukti Dhingra, Dipali G Sashital***

Roy J. Carver Department of Biochemistry, Biophysics, & Molecular Biology, Iowa State University, Ames, United States

**Abstract** CRISPR adaptation immunizes bacteria and archaea against viruses. During adaptation, the Cas1-Cas2 complex integrates fragments of invader DNA as spacers in the CRISPR array. Recently, an additional protein Cas4 has been implicated in selection and processing of prespacer substrates for Cas1-Cas2, although this mechanism remains unclear. We show that Cas4 interacts directly with Cas1-Cas2 forming a Cas4-Cas1-Cas2 complex that captures and processes prespacers prior to integration. Structural analysis of the Cas4-Cas1-Cas2 complex reveals two copies of Cas4 that closely interact with the two integrase active sites of Cas1, suggesting a mechanism for substrate handoff following processing. We also find that the Cas4-Cas1-Cas2 complex processes single-stranded DNA provided in cis or in trans with a double-stranded DNA duplex. Cas4 cleaves precisely upstream of PAM sequences, ensuring the acquisition of functional spacers. Our results explain how Cas4 cleavage coordinates with Cas1-Cas2 integration and defines the exact cleavage sites and specificity of Cas4.
DOI: https://doi.org/10.7554/eLife.44248.001

## Introduction

Bacteria and archaea use an adaptive immune system composed of clustered regularly interspaced short palindromic repeats (CRISPR) arrays and CRISPR-associated (Cas) proteins to defend against infection (*Barrangou et al., 2007*; *Brouns et al., 2008*; *Marraffini and Sontheimer, 2008*). Within this system, the CRISPR locus is programmed with 'spacer' sequences that are derived from foreign DNA and serve as a record of prior infection events (*Bolotin et al., 2005*; *Mojica et al., 2005*; *Pourcel et al., 2005*). The host adapts to an infection event when Cas proteins insert short fragments from the invader DNA as new spacers between repeating sequence elements within the CRISPR locus (reviewed in *Jackson et al., 2017*). The locus is transcribed and processed into short CRISPR RNAs (crRNAs), which assemble with Cas proteins to form a RNA-guided surveillance complex (reviewed in *Hochstrasser and Doudna, 2015*; *Charpentier et al., 2015*). Finally, the surveillance complex recognizes the target bearing complementarity to the crRNA sequence and a Cas nuclease cleaves or degrades the target during the interference stage (reviewed in *Marraffini, 2015*).

Although the machinery and mechanisms involved in CRISPR interference are extremely diverse (*Koonin et al., 2017*), the adaptation proteins Cas1 and Cas2 are conserved among most CRISPR systems, suggesting a common molecular mechanism for acquiring spacers. Cas1 and Cas2 form a heterohexameric complex that catalyzes spacer integration via two transesterification reactions mediated by nucleophilic attack of the 3'-hydroxyl on each strand of a double-stranded prespacer substrate at the phosphodiester backbone within the CRISPR array. Integration occurs at the first repeat in the CRISPR array, with one attack occurring between the upstream leader sequence and the repeat and the other occurring on the opposite strand between the repeat and first spacer

*For correspondence:
sashital@iastate.edu

Competing interests: The authors declare that no competing interests exist.

**eLife digest** Many people have now heard of CRISPR, or CRISPR-Cas9, as a gene editing technology. Yet CRISPR evolved in bacteria to protect them against viral infections. While parts of the CRISPR system are now being widely used, the research community still does not know everything about how the system operates in its natural setting.

In bacteria, CRISPR protects against infection by making lasting records of viruses a cell has encountered. It cuts short sections from the viral DNA and keeps them as a way to fight the virus if it ever returns. The key proteins in collecting and storing the virus DNA are called Cas1, Cas2 and Cas4. Previous work suggests that Cas4 is important for cutting suitable lengths of DNA for storage. Yet, how Cas4, Cas1 and Cas2 work together to select, cut and store DNA is not well studied.

Lee et al. have now used electron microscopy to examine how Cas1, Cas2 and Cas4 cooperate in the CRISPR system. The proteins studied came from bacteria called *Bacillus halodurans*. The structure revealed direct links between the Cas1 and Cas4 proteins that likely help to ensure these proteins are coordinated correctly to cut and store the DNA sections. Specifically, it showed that two Cas4 proteins interact with the two key active sites of Cas1. The findings also highlight that Cas4 cuts DNA at specific locations to make sure the resulting DNA sections are suitable for CRISPR protection.

The close association between Cas1 and Cas4 could be a critical aspect of the reliability of the CRISPR system in protecting bacteria from viruses. There are more bacteria on Earth than any other living thing. Understanding their biology has wide ranging environmental, health and bioengineering applications. In addition, learning more about the CRISPR system could further expand its potential to drive revolutionary biotechnology tools derived from these bacterial immune systems.

DOI: https://doi.org/10.7554/eLife.44248.002

within the array (*Arslan et al., 2014*; *Nuñez et al., 2015a*; *Rollie et al., 2015*). These reactions result in the insertion of the prespacer between two single-strand repeats, and this gapped intermediate is repaired by host factors (*Ivančić-Baće et al., 2015*).

In order to form a functional spacer, the adaptation complex must capture and process longer fragments of DNA from the invader containing a flanking sequence called a protospacer adjacent motif (PAM) (*Nuñez et al., 2015b*; *Wang et al., 2015*; *Xiao et al., 2017*). The PAM is an essential motif during target recognition by the surveillance complex and must be present next to the target in order for interference to occur (*Deveau et al., 2008*; *Redding et al., 2015*; *Sashital et al., 2012*; *Semenova et al., 2011*; *Sternberg et al., 2014*). However, the PAM is not part of the spacer and must be removed from the prespacer prior to integration through a processing step. In addition, integration must occur in the correct orientation to produce a crRNA that is complementary to the PAM-containing strand of the invader.

In some systems, additional Cas proteins, such as Cas4, are also required during adaptation. Cas4 is widespread in type I, II, and V systems (*Hudaiberdiev et al., 2017*). In in vivo studies, deletion of *cas4* reduced the adaptation efficiency (*Li et al., 2014*; *Liu et al., 2017*) and resulted in the acquisition of non-functional spacers from regions that lacked a correct PAM (*Almendros et al., 2019*; *Kieper et al., 2018*; *Shiimori et al., 2018*; *Zhang et al., 2019*). Some systems have two *cas4* genes that work together to define the PAM, length and orientation of spacers, suggesting that the two Cas4 proteins are involved in processing each end of the prespacer and that they may be present during integration (*Shiimori et al., 2018*). Similarly, in vitro studies have suggested that Cas4 is involved in PAM-dependent prespacer processing (*Lee et al., 2018*; *Rollie et al., 2018*). Cas4 endonucleolytically cleaves PAM-containing 3'-single-stranded overhangs that flank double-stranded prespacers (*Lee et al., 2018*). Importantly, Cas4 cleavage activity is dependent on the presence of Cas1 and Cas2, and Cas4 inhibits premature integration of unprocessed prespacers. These observations suggest that Cas4 associates with the Cas1-Cas2 complex, although direct biochemical and structural evidence for this Cas4-Cas1-Cas2 complex remains elusive (*Lee et al., 2018*; *Plagens et al., 2012*).

Here we show that Cas4 forms a complex with Cas1-Cas2 in the presence of dsDNA. Using single-particle negative-stain electron microscopy (EM), we determined the architecture of *Bacillus*

*halodurans* type I-C Cas1-Cas2 and Cas4-Cas1-Cas2 complexes. Unlike the symmetrical $Cas1_4$-$Cas2_2$ structure, we observed a mixture of symmetrical ($Cas4_2$-$Cas1_4$-$Cas2_2$) and asymmetrical ($Cas4_1$-$Cas1_4$-$Cas2_2$) complexes, suggesting a structural mechanism for distinguishing between the PAM and non-PAM end of the prespacer following processing. The positioning of Cas4 places it in close proximity to the Cas1 active site, suggesting a mechanism for substrate handoff following prespacer processing. Surprisingly, the Cas4-Cas1-Cas2 complex processes single-strand DNA when an activator duplex DNA is provided in trans. Using this ssDNA cleavage assay, we show that the Cas4-Cas1-Cas2 complex is highly specific for PAM sequences and cleaves precisely upstream of the PAM. In a duplex substrate, the PAM must be positioned within a single-strand region for optimal cleavage, but Cas4 can cleave at various locations within this single-stranded overhang. Collectively, these findings provide the first structural information of the Cas4-Cas1-Cas2 adaptation complex and reveal the precision and specificity of prespacer processing prior to integration.

## Results

### Formation of the Cas4-Cas1-Cas2 complex

We previously showed that *B. halodurans* type I-C Cas4 associates tightly with Cas1 but were unable to obtain the Cas4-Cas1-Cas2 complex due to instability of the Cas1-Cas2 complex in this system (*Lee et al., 2018*). We hypothesized that a CRISPR DNA substrate may help stabilize the complex. To test this possibility, we designed a CRISPR hairpin 'target' substrate containing a 10 bp leader, the full 32 bp repeat, and a 5 bp spacer, mimicking the region of the CRISPR at which integration occurs (*Figure 1A*). We incubated the individually purified Cas1 and Cas2 proteins with hairpin target DNA in equimolar amounts and removed unassociated DNA via ion-exchange chromatography followed by size-exclusion chromatography to remove free Cas proteins. Incubation of individual components with or without the hairpin target led to different elution volumes from a size-exclusion column (*Figure 1B*). In the absence of the target, Cas1 and Cas2 proteins generated two separate peaks, which correspond to the peaks of each individual component. When Cas1 and Cas2 were incubated with the target DNA, the proteins eluted earlier as a single peak, while unassociated Cas2 eluted at the original elution volume (*Figure 1B–C*). These data indicate that the Cas1-Cas2 complex from type I-C is stabilized in the presence of dsDNA.

Next, we attempted to reconstitute the putative Cas4-Cas1-Cas2 complex in the presence of the CRISPR hairpin target (*Figure 1A*). Following incubation of equimolar amounts of each component, Cas4, Cas1 and Cas2 eluted in a single peak from the size-exclusion column (*Figure 1—figure supplement 1*), with an earlier elution volume than the Cas1-Cas2-target sample (*Figure 1B–C*). In addition, we observed two peaks with the approximate elution volumes of free Cas1 and Cas2 (*Figure 1—figure supplement 1*). Both Cas1-Cas2 and Cas4-Cas1-Cas2 complexes contained the hairpin target DNA (*Figure 1D*). Together, these data suggest that the proteins directly interact with the hairpin target, and the formation of the higher-order complex is also stabilized by the presence of dsDNA substrates.

### Architecture of the Cas4-Cas1-Cas2 complex

To characterize the molecular architecture of the complexes, we next performed single-particle electron microscopy (EM) of negatively stained Cas1-Cas2 or Cas4-Cas1-Cas2 complexes bound to the target. Raw micrographs and two-dimensional (2D) class averages revealed particles with fairly homogenous size and symmetrical architecture consistent with the known structure of the Cas1-Cas2 complex (*Nuñez et al., 2014*; *Nuñez et al., 2015b*; *Wang et al., 2015*; *Xiao et al., 2017*) (*Figure 2—figure supplement 1A–B*). Some 2D class averages for the Cas4-Cas1-Cas2 complex contained clear additional density, suggesting the presence of ordered Cas4 within the complex (*Figure 1E–F*).

For the Cas1-Cas2 complex, we determined a 22 Å three-dimensional (3D) reconstruction (*Figure 2A*). The EM density revealed clear C2 symmetry, which was enforced during the final 3D refinement (*Figure 2—figure supplement 2A*). Segmentation of the density revealed three clear domains, corresponding to two Cas1 dimers sandwiching a Cas2 dimer (*Figure 2A*). We fit the *B. halodurans* Cas2 structure and a structural model of *B. halodurans* Cas1 (see Materials and methods) to the segmented densities. The structural models fit well, and the overall architecture of the

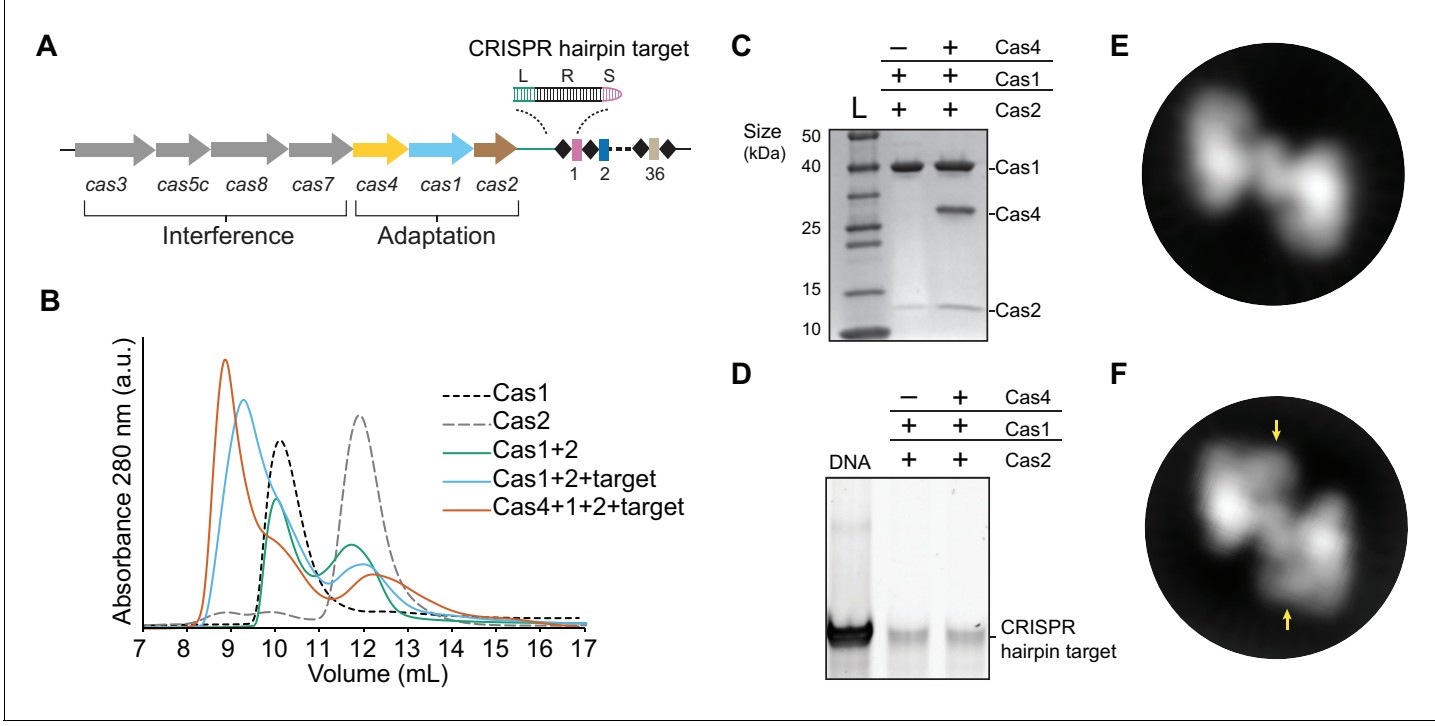

**Figure 1.** Complex formation of *B. halodurans* Cas1-Cas2 or Cas4-Cas1-Cas2 in the presence of CRISPR hairpin target. (**A**) Overview of the *cas* genes and CRISPR locus found in the *Bacillus halodurans* type I-C system. Spacers are shown as rectangles, repeats are shown as diamonds, each *cas* gene is shown as an arrow and gene products involved in adaptation or interference are indicated. The CRISPR hairpin target used for this study contains a 10 bp leader (L, green), the full 32 bp repeat (R, black), and a 5 bp spacer (S, purple). (**B**) Size-exclusion chromatography (SEC) of various combinations of Cas1, Cas2, Cas4 and target DNA. (**C**) Coomassie-blue stained SDS-PAGE gel of proteins present in the earliest eluting peak fractions of SEC following complex formation. (**D**) SYBR Gold stained 10% PAGE gel of DNA present in the earliest eluting peak fractions of SEC following complex formation. (**E**) Representative 2D class average of the Cas1-Cas2 complex. (**F**) Representative 2D class average of the Cas4-Cas1-Cas2 complex. Extra density corresponding to Cas4 is indicated by arrows.

DOI: https://doi.org/10.7554/eLife.44248.003

The following figure supplements are available for figure 1:

**Figure supplement 1.** Analysis of Cas4-Cas1-Cas2 formation.

DOI: https://doi.org/10.7554/eLife.44248.004

**Figure supplement 2.** Formation of the Cas4-Cas1-Cas2 complex in the presence of prespacer DNA.

DOI: https://doi.org/10.7554/eLife.44248.005

complex was similar to that of the crystal structure of the type II-A Cas1-Cas2-prespacer complex from *Enterococcus faecalis* (*Figure 2—figure supplement 3A,B*). In contrast, the *E. coli* type I-E Cas1-Cas2-prespacer complex (*Figure 2—figure supplement 3C*) did not fit well in the Cas1-Cas2 density, revealing that the architecture of type I-C Cas1-Cas2 may be more similar to type II-A than to another type I system (*Nuñez et al., 2015b*; *Xiao et al., 2017*).

For the Cas4-Cas1-Cas2 complex, we determined a 20 Å 3D reconstruction of symmetrical particles enforcing C2 symmetry (*Figure 2—figure supplement 2B*). The segmented density clearly reveals the base Cas1-Cas2 architecture, along with additional density corresponding to two molecules of Cas4 (*Figure 2B*). During 3D classification of particles, we observed two classes containing approximately 50% of Cas4-Cas1-Cas2 particles that appeared to contain density for only a single Cas4 molecule (*Figure 2—figure supplement 2B*). A subset of these particles was refined as a separate 21 Å 3D reconstruction without symmetry enforced, revealing an asymmetrical Cas4-Cas1-Cas2 complex with 1:4:2 stoichiometry (*Figure 2C*). These particles may represent ternary complexes in a partially dissociated state, or incomplete formation of the 2:4:2 stoichiometry complex due to reduced affinity for Cas4 within the ternary complex.

In both the symmetrical and asymmetrical Cas4-Cas1-Cas2 reconstructions, the Cas4 and Cas1 densities are contiguous but Cas4 appears to be distinct from the Cas2 density (*Figure 2B–C*). This

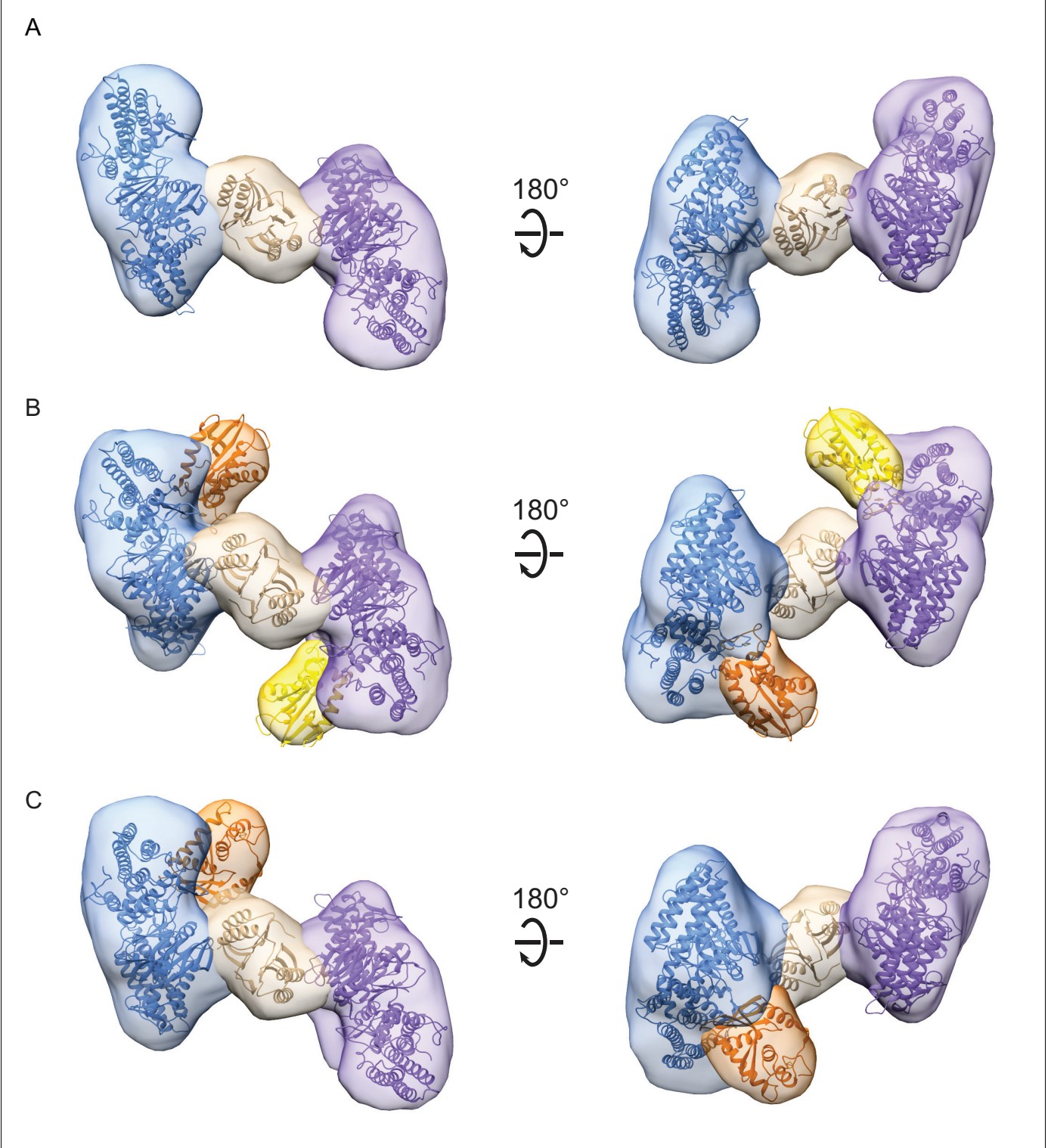

**Figure 2.** Architecture of Cas1-Cas2 and Cas4-Cas1-Cas2 complexes formed in the presence of CRISPR hairpin target DNA. (A) Segmented density for Cas1-Cas2 reconstruction. Two copies of a structural model of BhCas1 dimer (see Materials and methods) were fit in the two assigned Cas1 densities (blue, purple). The crystal structure of BhCas2 (PDB 4ES3) was used for fitting to density assigned to Cas2 (tan) (*Nam et al., 2012*). (B–C) Segmented density for (B) symmetrical and (C) asymmetrical reconstructions of Cas4-Cas1-Cas2. BhCas1 and BhCas2 structural models are fit to segments and colored as in (A). Two copies of a structural model of BhCas4 are fit into assigned Cas4 densities in (B) (orange, gold). One copy of BhCas4 structural model is fit into assigned Cas4 density in (C) (orange).

*Figure 2 continued on next page*

*Figure 2 continued*

DOI: https://doi.org/10.7554/eLife.44248.006

The following figure supplements are available for figure 2:

**Figure supplement 1.** Single particle EM analysis of Cas1-Cas2 and Cas4-Cas1-Cas2.

DOI: https://doi.org/10.7554/eLife.44248.007

**Figure supplement 2.** Three-dimensional classification of Cas1-Cas2 and Cas4-Cas1-Cas2.

DOI: https://doi.org/10.7554/eLife.44248.008

**Figure supplement 3.** Modeling of Cas1-Cas2 structures.

DOI: https://doi.org/10.7554/eLife.44248.009

**Figure supplement 4.** Modelling possible orientations of Cas4 within assigned density.

DOI: https://doi.org/10.7554/eLife.44248.010

**Figure supplement 5.** Model of Cas1-Cas2 full-site integration product fit in symmetrical Cas4-Cas1-Cas2-target reconstruction.

DOI: https://doi.org/10.7554/eLife.44248.011

**Figure supplement 6.** Comparison of Cas4-Cas1-Cas2 complexes formed in the presence of CRISPR hairpin target or prespacer DNA.

DOI: https://doi.org/10.7554/eLife.44248.012

suggests that Cas4 interacts with Cas1 but not Cas2 within the ternary complex, similar to the tight interaction we have previously observed between Cas4 and Cas1 in the absence of Cas2 (*Lee et al., 2018*). However, the interaction surface between Cas4-Cas1 appears to be different in the context of the binary and ternary complexes, suggesting that Cas4 and Cas1 may have two alternative modes of interaction.

A high-resolution structure of the BhCas4 sequence is not available, and the closest homolog from *Pyrobaculum calidifontis* (PDB 4R5Q) contains a long N-terminal domain that is likely not present in the BhCas4 structure (*Figure 2—figure supplement 4A*). Therefore, for modeling Cas4 into the Cas4-Cas1-Cas2 density, we used a predicted structural model (see Materials and methods) that reflects the putatively more compact BhCas4 structure (*Figure 2—figure supplement 4B*). Fitting this structural model into the segmented Cas4 density revealed four possible orientations of the protein with respect to Cas1 (*Figure 2—figure supplement 4C–E*). The segmented density for Cas4 is smaller than the structure and parts of the structure fit into the Cas1 density, suggesting that the segmentation between Cas1 and Cas4 densities was incomplete due to low resolution as has been observed previously (*Pintilie et al., 2010*). All four orientations position the Cas4 active site with varying degrees of proximity (19–38 Å) to the Cas1 active sites that bind the 3'-OH end of the prespacer in structures of Cas1-Cas2-prespacer (*Figure 2—figure supplement 4D–E*). These active sites act as integrases for the two steps of integration. The proximity between Cas4 and the Cas1 active site suggests that the product of Cas4 cleavage could transit to the integrase active sites following cleavage.

The Cas1-Cas2 and Cas4-Cas1-Cas2 EM volumes did not contain any obvious cylindrical density corresponding to the hairpin target DNA, consistent with prior observations that DNA is not readily observable by negative-stain single-particle EM (*Hochstrasser et al., 2014*; *Nogales and Scheres, 2015*). We also note that the Cas4 density lies along the same surface of Cas1 that is expected to interact with elements within the CRISPR array during half-site and full-site integration, based on crystal structures of these intermediates (*Figure 2—figure supplement 5*). This suggests that the CRISPR hairpin target is either not present in the complex, or that it is bound in an alternate location. The observation that DNA co-elutes with the complex from the size-exclusion column disfavors the former possibility (*Figure 1D*). Notably, in the highest-resolution symmetrical Cas4-Cas1-Cas2 reconstruction, we observe additional density along the prespacer-binding surface of Cas2, but not the repeat-binding surface (*Figure 2—figure supplement 5* middle). Although we cannot definitively define this density as DNA, this observation suggests the DNA may be bound on the prespacer side of the complex and that the CRISPR target in our complex is not bound along the same surface as in prior crystal structures. Together, these observations suggest that Cas4 binding may be mutually exclusive with CRISPR binding along the repeat-binding surface, and that Cas4 may dissociate prior to integration.

## Formation of the Cas4-Cas1-Cas2 complex with prespacer substrate

Although we hypothesized that the CRISPR hairpin target would stabilize the Cas1-Cas2 complex by binding along the repeat-binding surface, our structural studies of complexes formed in the presence of the hairpin target suggest this is not the case. Notably, the prespacer-binding surface of Cas2 interacts with DNA through non-specific backbone contacts (*Nuñez et al., 2015b*; *Wang et al., 2015*; *Xiao et al., 2017*), and no structure of Cas1-Cas2 bound to CRISPR DNA alone has been previously reported. This could suggest that any dsDNA, including the CRISPR target, may preferentially bind along the prespacer-binding surface and that other DNA substrates binding in that location could promote stable complex formation. We therefore attempted to form the Cas4-Cas1-Cas2 complex using a prespacer substrate containing a 24 bp duplex with 15-nt 3' overhangs containing 5'-GAA-3' PAM sequences, which we previously showed is processed by Cas4 in the presence of Cas1 and Cas2 (*Lee et al., 2018*). Using a modified protocol (see Materials and methods), we were able to successfully purify a peak containing Cas4, Cas1, Cas2, and prespacer DNA from a size-exclusion column (*Figure 1—figure supplement 2*). The peak eluted slightly later than the Cas4-Cas1-Cas2-target complex, likely due to the smaller size of the DNA substrate (*Figure 1—figure supplement 2A*). As with the Cas4-Cas1-Cas2-target complex, all three proteins and DNA were present in the peak fractions (*Figure 1—figure supplement 2B–C*).

To determine whether the prespacer complex has a similar architecture to the complex bound to the CRISPR hairpin target, we performed negative stain EM analysis of Cas4-Cas1-Cas2-prespacer (*Figure 2—figure supplements 1*, *2* and *6*). The 2D class averages revealed fewer symmetrical classes (*Figure 2—figure supplement 1B*), and 3D classification revealed a smaller proportion of particles containing two copies of Cas4 relative to particles containing one or no copies of Cas4 (*Figure 2—figure supplement 2C*). These data suggest that Cas4 is less stably bound in the Cas4-Cas1-Cas2-prespacer complex than in a complex containing a longer dsDNA. Both EM densities were at a lower resolution than the complexes solved using the hairpin target (symmetrical EM density 22 Å, asymmetrical EM density 24 Å). Although features are less well-defined for these lower resolution densities, the overall architecture is similar between the complexes containing the two different DNA substrates (*Figure 2—figure supplement 6*), indicating that complex formation occurs similarly regardless of the type of DNA substrate.

## Cas4 is activated for ssDNA processing in the presence of dsDNA

The observation that the presence of dsDNA substrates stabilized the Cas4-Cas1-Cas2 complex prompted us to hypothesize that binding to the CRISPR DNA may enhance processing activity by stimulating complex formation. To test this hypothesis, we analyzed cleavage of the prespacer substrate containing a 24 bp duplex with 15-nt 3' overhangs containing 5'-GAA-3' PAM sequences, in the absence or presence of CRISPR DNA (*Figure 3A*). In addition, we tested cleavage of a single 39-nt strand of this substrate (*Figure 3A*). Interestingly, for the ssDNA substrate, we observed cleavage product in the presence of the CRISPR DNA, while the DNA remained uncleaved in the absence of the CRISPR DNA or Cas4 (*Figure 3B*). However, we observed similar amounts of cleavage for duplex prespacers with and without the CRISPR, suggesting that the presence of the CRISPR DNA only enhances cleavage of the ssDNA substrate (*Figure 3B*). Importantly, we previously showed that cleavage of the duplex prespacer is dependent on having a PAM present in the single-stranded overhang, indicating that cleavage occurs in a PAM-dependent manner (*Lee et al., 2018*). Consistently, both the ssDNA substrate and the duplex substrate produced the same length product, suggesting they were cleaved at the PAM site located within the 3'-overhang of the duplex substrate. These data suggest that Cas4-Cas1-Cas2 can cleave PAM sites in ssDNA when a CRISPR DNA is provided in trans, and that the dsDNA region present within the duplex prespacer is sufficient to stimulate complex formation.

We previously observed that Cas4 cleavage products can be integrated into mini-CRISPR arrays (*Lee et al., 2018*) (*Figure 3—figure supplement 1A*). Notably, although the cleavage product for the duplex prespacer was integrated into the CRISPR (*Figure 3B*), no integration product was visible for the single-stranded substrate following processing even at increased image contrast (*Figure 3—figure supplement 1B*). In the absence of Cas4, a small amount of unprocessed integration product was visible when the image contrast was increased for both the ssDNA and duplex prespacer. Together, these results suggest that although Cas1 can integrate ssDNA, optimal integration

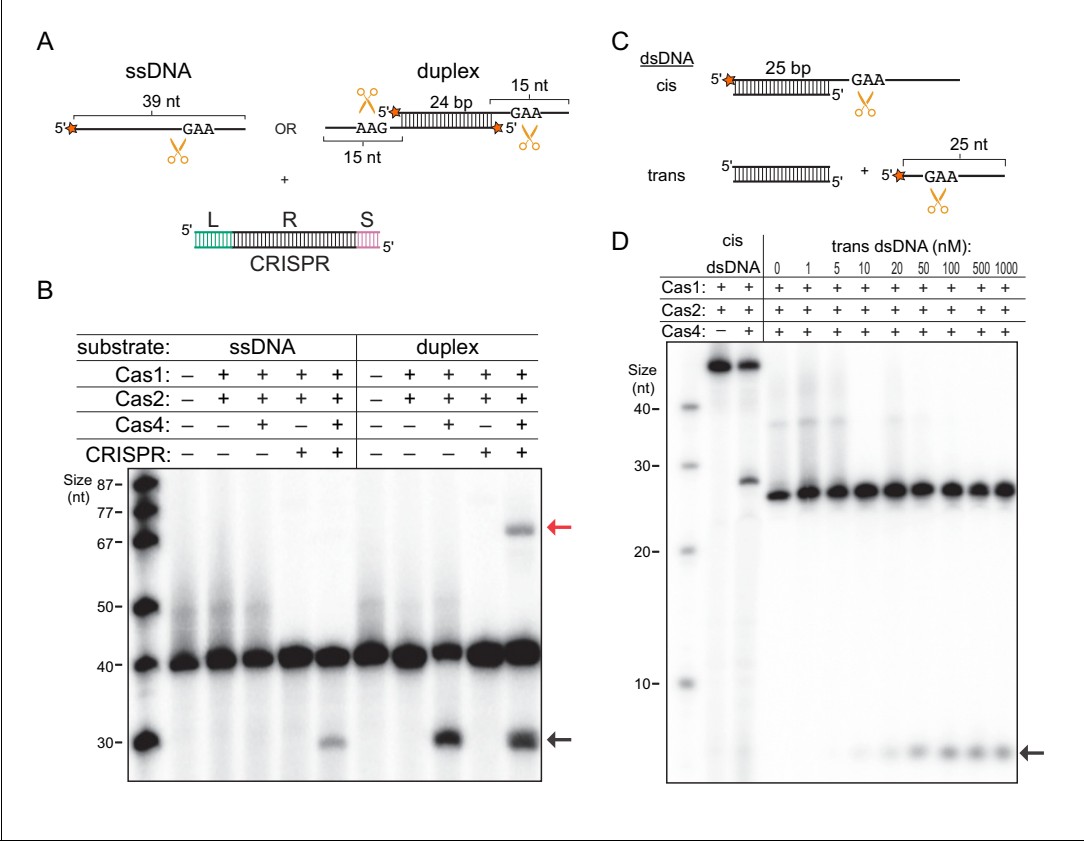

**Figure 3.** Single-stranded DNA processing by the Cas4-Cas1-Cas2 complex. (**A**) Schematic of prespacer cleavage assay for (**B**). L indicates leader, R indicates repeat, S indicates spacer in the CRISPR DNA substrate. Radiolabel is indicated with a star. (**B**) Prespacer processing assay using ssDNA or duplex prespacer in the absence or presence of CRISPR DNA. Black arrow indicates the cleavage product. Red arrow indicates integration products following processing. (**C**) Schematic of cleavage assay using 25 nt single-stranded substrates provided in cis or in trans with a 25 bp duplex. (**D**) Cleavage assay using cis dsDNA or 25-nt ssDNA with titration of 25 bp duplex provided in trans.

DOI: https://doi.org/10.7554/eLife.44248.013

The following figure supplement is available for figure 3:

**Figure supplement 1.** Integration of unprocessed and processed prespacers.

DOI: https://doi.org/10.7554/eLife.44248.014

requires that the single-stranded region be attached to a duplex, which may facilitate handoff of the processed end from Cas4 to the Cas1 active site.

Our results suggest that any dsDNA, not just the CRISPR DNA, may enhance ssDNA cleavage activity either when present in cis with the ssDNA region or when provided in trans. To test these possibilities, we performed a cleavage assay with two sets of substrates: a 25 bp duplex with one 25-nt 3' overhang or 5'-end-$^{32}$P-labeled 25-nt ssDNA with an unlabeled blunt-end 25 bp duplex added in trans (*Figure 3C*). As expected, the duplex substrate was cleaved within the single-stranded overhang region in the presence of Cas4, similar to the duplex substrate tested in *Figure 3B* (*Figure 3D*). For the ssDNA substrate, no cleavage was observed in the absence or at low concentrations of dsDNA, while cleavage product accumulated at higher dsDNA concentrations (*Figure 3D*). A similar amount of cleavage product was observed at dsDNA concentrations ≥ 100 nM, which is higher than the expected concentration of the Cas4-Cas1-Cas2 complex (50 nM). These results are consistent with our model that dsDNA directly stabilizes the complex, and that excess dsDNA should not induce additional activation. Overall, our results show that type I-C Cas4-Cas1-Cas2 adaptation complex can be stabilized by dsDNA and that, within this complex, Cas4 is activated for ssDNA cleavage for substrate provided either in cis or in trans.

## Precise PAM-specific DNA processing by Cas4-Cas1-Cas2

We previously showed that Cas4 processes prespacers in a PAM-dependent manner (*Lee et al., 2018*), but the exact cleavage sites and specificity of Cas4 remain unclear. The observation that Cas4-Cas1-Cas2 can process ssDNA allowed us to more precisely define the cleavage site. By using ssDNA substrates, we could more readily test sequences containing multiple PAM sites or a variety of sequences, and additionally include primer-binding sites for Sanger sequencing reactions. We first conducted prespacer processing assays using ssDNA substrates containing a 5'-GAA-3' PAM between T-rich sequences in the presence of activating dsDNA. Comparison with ddNTP Sanger sequencing reactions revealed that Cas4-Cas1-Cas2 precisely cleaved the ssDNA directly upstream of the PAM, while Cas1-Cas2 or Cas4 alone failed to cleave the substrates (*Figure 4A*). This cleavage site is consistent with the expected processing site relative to the PAM required to form a functional spacer during spacer acquisition.

We next tested whether the Cas4-Cas1-Cas2 could cleave at multiple PAM sites within a single substrate, and whether location of the PAM sites affects the processing activity. We designed ssDNA substrates containing three PAM sites at varying intervals. Substrates with 10, 8, 6, 4 or 2-nt between three PAM sites generated three different sized cleavage products, resulting from cleavage directly upstream of each PAM site (*Figure 4B* and *Figure 4—figure supplement 1*). However, when we introduced three PAM sites consecutively on ssDNA substrates, we observed a predominant product at the first PAM position (*Figure 4C*), indicating that processing was inhibited at the second and

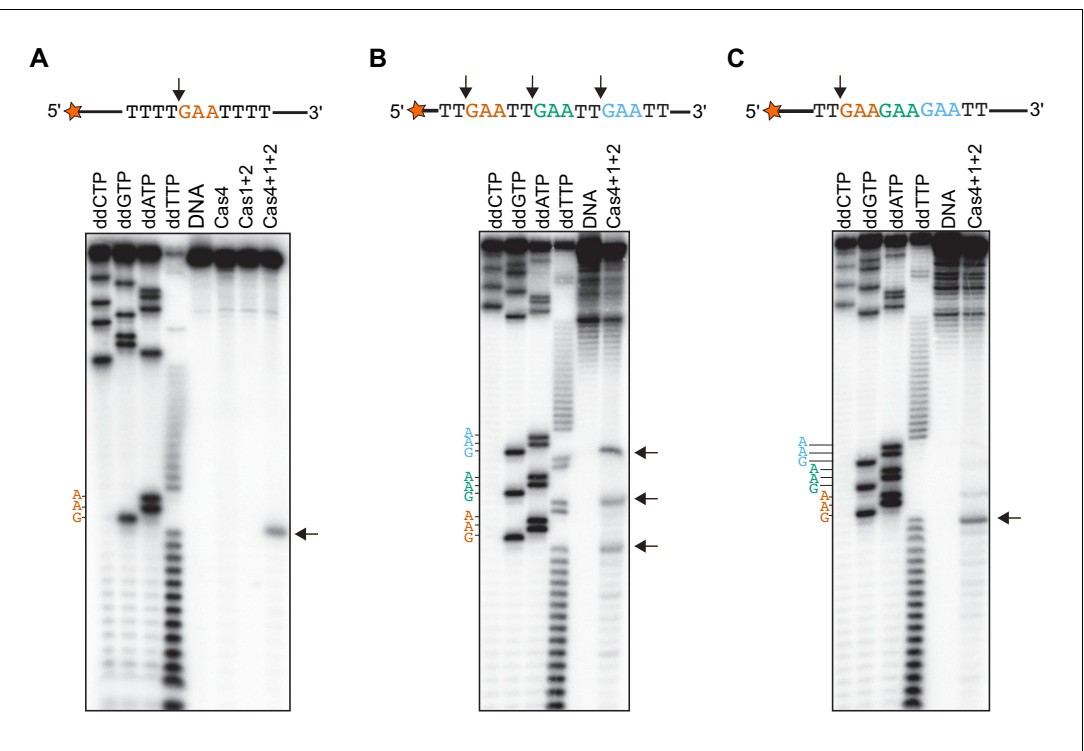

**Figure 4.** Cas4-Cas1-Cas2 processes directly upstream of PAM sites. (**A**) Prespacer processing assay for ssDNA containing one PAM (GAA) site between T-rich sequences. (**B**) Prespacer processing assay for ssDNA containing three PAM sites with 2-nt intervals. (**C**) Prespacer processing assay with ssDNA containing three consecutive PAM sites. The first four lanes are Sanger sequencing reactions using the indicated ddNTP. The lane labeled DNA is a negative control reaction in which no proteins were added. Lanes labeled 'Cas4', 'Cas1 + 2' or 'Cas4 + 1 + 2' are reactions performed with the indicated proteins. Arrows indicate the predominant cleavage site.

DOI: https://doi.org/10.7554/eLife.44248.015

The following figure supplement is available for figure 4:

**Figure supplement 1.** Cleavage of ssDNA with three PAMs.

DOI: https://doi.org/10.7554/eLife.44248.016

third PAM site. Together, these results suggest that the adaptation complex can process multiple PAM sites within a single substrate as long as the PAMs are not consecutive.

To explore whether the PAM-flanking sequences affect cleavage and whether cleavage could be observed at non-PAM sequences, we used PAM-flanking sequences containing either A-T rich or 'random' sequences in which all four nucleotides were represented (*Figure 5*). Although we observed low levels of non-specific degradation in the presence of Cas4-Cas1-Cas2, all substrates displayed a single predominant cleavage product directly upstream of the PAM site (*Figure 5A–D*). These results indicate that Cas4 is highly PAM specific, as it did not cleave efficiently at other sites in the random sequences.

We observed relatively low cleavage for substrates containing random sequences upstream of the PAM, suggesting that the PAM-flanking sequence may have an effect on cleavage efficiency. To test this possibility, we used ssDNA substrates containing degenerate PAM-flanking sequences for Cas4-Cas1-Cas2 cleavage (*Figure 5—figure supplement 1A–B*). Using a PCR strategy that selectively amplified the uncleaved fraction (*Figure 5—figure supplement 1A,C*), we analyzed the relative depletion of flanking sequence in the absence or presence of Cas4 by high-throughput sequencing. We detected no significant differences in relative sequence levels for the –Cas4 and +Cas4 samples (*Figure 5—figure supplement 1D–E*), indicating that all PAM-flanking sequences were cleaved with the same efficiency. Together, our cleavage analyses strongly suggest that Cas4 cleavage specificity is dictated only by the presence of a PAM sequence within a single-stranded region of the substrate.

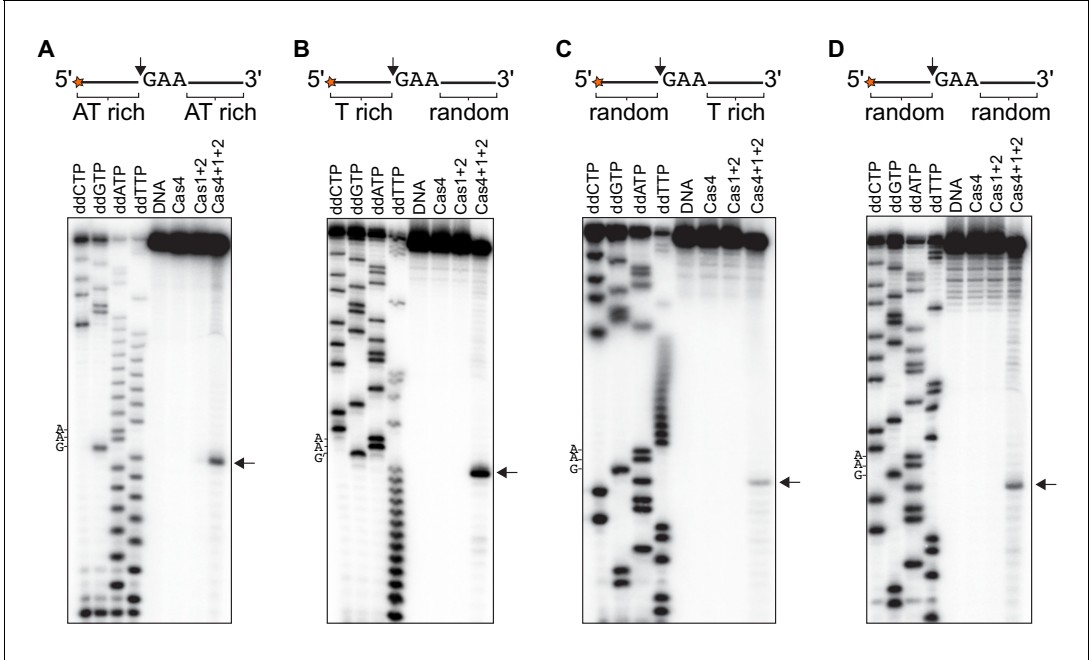

**Figure 5.** Cleavage of ssDNA substrates with different PAM-flanking regions. (**A**) Substrate with AT-rich sequences upstream and downstream of the PAM. (**B**) Substrate with T-rich sequence upstream and random sequence downstream of PAM. (**C**) Substrate with random sequence upstream and T-rich sequence downstream of PAM. (**D**) Substrate with random sequence upstream and downstream of PAM. The first four lanes are Sanger sequencing reactions using the indicated ddNTP. The lane labeled DNA is a negative control reaction in which no proteins were added. Lanes labeled 'Cas4', 'Cas1 + 2' or 'Cas4 + 1 + 2' are reactions performed with the indicated proteins. Arrow indicates predominant cleavage site.
DOI: https://doi.org/10.7554/eLife.44248.017

The following figure supplement is available for figure 5:

**Figure supplement 1.** PAM-flanking sequence depletion assay.
DOI: https://doi.org/10.7554/eLife.44248.018

## Cas4 cleavage depends on PAM location within single-stranded overhangs

Our data strongly suggests that Cas4 can cleave PAM sites within single-stranded DNA substrates, and that cleavage occurs at the same site when the ssDNA is provided in cis or in trans with the activating dsDNA (*Figure 3B*). However, for a duplex substrate containing a ssDNA overhang, it is also possible that Cas1-Cas2-substrate binding positions the single-stranded overhangs into the Cas4 active site. In that case, Cas4 could cleave the DNA based on a ruler mechanism, with the cleavage site defined by the distance between the end of the duplex and the Cas4 active site. To test this possibility, we designed a panel of duplex substrates in which the PAM position was varied within the single-strand overhang (*Figure 6A*). The four different substrates yielded products of different lengths, indicating that they were cleaved based on the position of the PAM within the single-stranded overhang (*Figure 6B*). Interestingly, Cas4 processed these substrates to varying degrees (*Figure 6B–C*). The substrate in which the PAM was located close to the duplex (two nt between

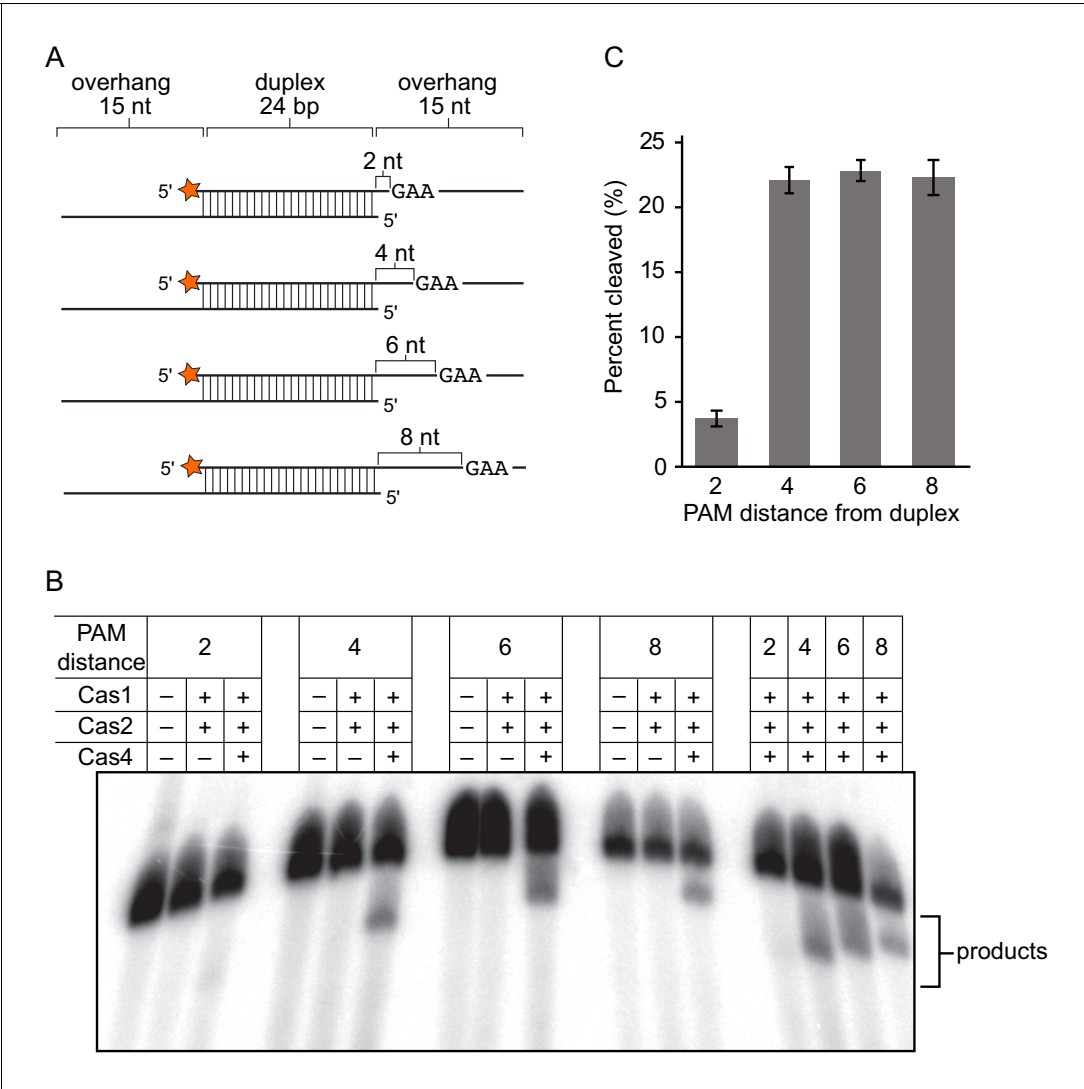

**Figure 6.** Processing of duplex prespacers with varied PAM positions in single-strand overhangs. (A) Panel of substrates used in processing experiments. The 5'-GAA-3' PAM begins after 2, 4, 6 or eight nt from the end of the duplex. Radiolabel is indicated with an orange star. (B) Polyacrylamide gel image showing cleavage of substrates shown in (A). A second set of +Cas4 reactions were loaded in the last four lanes on the right for ease of comparison of product sizes. (C) Quantitation of percent cleaved for substrates shown in (A). The average of three replicates is shown, with error bars representing standard deviation.
DOI: https://doi.org/10.7554/eLife.44248.019

duplex and PAM site) was cleaved inefficiently, while the other three substrates were cleaved with similar efficiency (*Figure 6C*). These data, along with prior results showing that Cas4 cannot cleave a PAM contained within a dsDNA (*Lee et al., 2018*), suggest that the PAM must be located at a longer distance from the duplex to be accommodated within the Cas4 active site.

## Discussion

Cas4 is a core family of CRISPR adaptation proteins, but its exact mechanism in spacer acquisition is relatively poorly understood. In particular, although there has been some preliminary biochemical evidence that Cas4 directly associates with Cas1-Cas2 to form a higher-order complex, these complexes were either very weak (*Lee et al., 2018*) or formed only under renaturing conditions (*Plagens et al., 2012*). Here, we discovered that the presence of dsDNA substrates stabilizes the formation of both Cas1-Cas2 and Cas4-Cas1-Cas2 complexes in the *B. halodurans* type I-C system. For the first time, we present the architecture of the Cas4-Cas1-Cas2 complex that mediates prespacer selection, processing, and integration during CRISPR adaptation.

Our structural analysis of *B. halodurans* type I-C adaptation complexes reveals a structure that is mutually exclusive with the previously determined Cas4-Cas1 complex (*Lee et al., 2018*). In the Cas4-Cas1 complex, the two Cas1 dimers are in close proximity and would exclude the Cas2 dimer. In addition, the interaction surface between Cas1 and Cas4 appears different in the two complexes. In Cas4-Cas1, the two Cas4 molecules each interact with one wing tip of the butterfly-like Cas1 dimers. In the Cas4-Cas1-Cas2 complex, Cas4 appears to interact along the length of one Cas1 wing. These results strongly suggest that Cas4-Cas1 must fully dissociate in order for the Cas1-Cas2 and Cas4-Cas1-Cas2 complex to form. Interestingly, some systems contain Cas4-1 fusion proteins (*Hudaiberdiev et al., 2017*), and a recent study indicates that the Cas4 domain performs a similar function in prespacer processing within this fusion (*Almendros et al., 2019*). Complexes formed with Cas4-1 fusions would likely have altered stoichiometry from that observed in our study. The impact of this altered stoichiometry will be an intriguing line of future inquiry.

We previously demonstrated that Cas4 inhibits integration of unprocessed prespacers by Cas1-Cas2 (*Lee et al., 2018*). These results suggested that Cas4 sequesters 3'-overhangs of the prespacers away from the Cas1 active site prior to processing. Our structural analysis of the Cas4-Cas1-Cas2 complex provides further evidence toward this model. Fitting of structural models into the Cas4-Cas1-Cas2 reconstruction suggests that Cas4 and Cas1 interact extensively and that their active sites may be in close proximity. Thus, it is possible that Cas4 blocks binding of the ssDNA in the Cas1 active site prior to processing. Other RecB-like nucleases bind single-stranded DNA by threading the substrate through a hole formed in the donut-shaped nuclease domain (*Zhou et al., 2015*). Similar binding by Cas4 may prevent release of the ssDNA until cleavage, at which point the upstream product would be released, allowing handoff to the Cas1 active site and enabling integration.

Previously, we showed that Cas4 processes prespacers with duplexes flanked by 3' overhangs, and that processing activity was independent of the duplex or overhang lengths (*Lee et al., 2018*). Our current results reveal that the duplex region of these prespacers likely activates Cas4-Cas1-Cas2 for cleavage, and that a similar processing activity can be activated when the duplex and ssDNA are provided in trans. Importantly, single-stranded substrates were processed less efficiently and were not integrated into the CRISPR array when provided in trans with the duplex, while single-strand overhangs of duplex substrates were integrated following processing. These results suggest that interactions of the prespacer duplex along the length of the Cas1-Cas2 core anchors the substrate to the complex, facilitating efficient Cas4 cleavage and direct handoff of the 3' overhangs from Cas4 to Cas1. In the absence of the cis duplex, the ssDNA is likely to be released by the complex upon Cas4 cleavage due to the lack of an anchoring interaction with the complex. Therefore, a duplex DNA containing single-strand overhangs is likely the optimal Cas4-Cas1-Cas2 substrate to enable efficient cleavage and substrate handoff. It remains unclear how these types of prespacers are formed. In other CRISPR-Cas sub-types, Cas4 has been demonstrated to have exonucleolytic activity that can lead to dsDNA unwinding (*Lemak et al., 2013*; *Lemak et al., 2014*; *Zhang et al., 2012*), although we have not observed robust exonuclease or dsDNA cleavage activity for the type I-C BhCas4 (*Lee et al., 2018*). It is possible that other host factors or conditions are required to unwind

ends of dsDNA, or to expose single-stranded overhangs for precise cleavage by Cas4 in the type I-C system.

Surprisingly, Cas4 processing can occur at several locations within both trans and cis ssDNA, as long as enough space is available between cleavage sites and the end of a duplex. Importantly, we do not observe Cas4-dependent PAM processing in the absence of Cas1-Cas2. These observations suggest that the Cas4 active site is activated for cleavage based on interactions with Cas1-Cas2. Once activated, Cas4 can sample any region of a single-stranded DNA that is accessible to the active site. This feature could allow more flexibility in defining prespacer substrates due to the ability of Cas4 to search for PAM sequences within single-stranded overhangs.

Previous in vitro studies have suggested that Cas4 processing is PAM-specific, although the exact position of processing was unclear (*Lee et al., 2018*; *Rollie et al., 2018*). We find that processing is highly PAM-specific and occurs precisely upstream of the PAM, consistent with the expected processing position to form a functional prespacer. In these experiments, Cas4 did not appear to have strong cleavage activity at non-PAM sites, suggesting that Cas4-Cas1-Cas2 only processes the PAM-proximal end in type I-C. In type I-A from *Pyrococcus furiousus*, two distinct Cas4 proteins coordinate the processing of each end of the prespacer (*Shiimori et al., 2018*). However, most Cas4-containing systems, including type I-C, lack a second *cas4* gene. It is possible that in these systems, Cas4 may define the PAM-distal end of the prespacer through an alternative cleavage activity or that another host factor is required for this processing activity.

Formation of functional spacers requires that the PAM end of the prespacer must be integrated at the leader-distal end of the repeat following prespacer processing. The precise details of how spacer orientation is defined during integration remain unknown. In type I-E, one nucleotide of the PAM is retained following prespacer processing, and this nucleotide may help to define the PAM end during integration (*Datsenko et al., 2012*; *Swarts et al., 2012*). In the type I-A system, the two Cas4 proteins are required to define orientation (*Shiimori et al., 2018*), suggesting that their presence in an adaptation complex may define the orientation of the spacer after the prespacer is fully processed. Notably, we observed asymmetrical complexes of Cas4-Cas1-Cas2 containing only one Cas4 subunit. This configuration, along with the hypothesis that Cas4 only processes the PAM end of the prespacer, could suggest that only a single copy of Cas4 is required to form a functional adaptation complex. Indeed, the asymmetrical complex may also define prespacer orientation, based on which end of the prespacer is bound at the Cas4-end of the complex. Our structural model also indicates that Cas4 may be mutually exclusive with binding at the CRISPR array (*Figure 2—figure supplement 5*), suggesting that the end of the asymmetrical complex lacking Cas4 may preferentially bind to the CRISPR array. We previously showed that integration occurs first at the leader-end of the repeat (*Lee et al., 2018*), which is also the site of integration for the non-PAM end of the prespacer. Thus, it is possible that an asymmetrical complex may define spacer orientation by ensuring that the non-PAM end is integrated first. Future studies will be required to determine the timing of prespacer processing, CRISPR binding and whether or not Cas4 dissociation is required prior to integration.

## Data availability

The negative-stain EM volumes for the Cas1-Cas2-target, asymmetrical Cas4-Cas1-Cas2-target, symmetrical Cas4-Cas1-Cas2-target, asymmetrical Cas4-Cas1-Cas2-prespacer and symmetrical Cas4-Cas1-Cas2-prespacer complexes have been deposited to EMDB under the accession numbers EMDB-20127, EMDB-20128, EMDB-20129, EMDB-20130 and EMDB-20131, respectively.

## Contact for reagent and resource sharing

Further information and requests for resources and reagents should be directed to and will be fulfilled by Dipali Sashital (sashital@iastate.edu).

## Experimental model and subject details

### *Escherichia coli* BL21 (DE3)

*E. coli* BL21 (DE3) cells were used for protein production of Cas1 and Cas2. Cells were grown at 16°C in LB medium supplemented with 50 µg/mL kanamycin.

*Escherichia coli* BL21 Star (DE3)

*E. coli* BL21 Star (DE3) cells were used for protein production of Cas4 with pRKSUF017 for in vitro experiments. Cells were grown at 18°C in 2xYT medium supplemented with 25 µg/mL carbenicillin and 2.5 µg/mL tetracycline.

## Materials and methods

Key resources table

| Reagent type (species) or resource | Designation | Source or reference | Identifiers | Additional information |
|---|---|---|---|---|
| Strain, strain background (*Escherichia coli*) | BL21 Star (DE3) | Thermo Fisher Scientific | C6010-03 | |
| Strain, strain background (*Escherichia coli*) | BL21 (DE3) | New England Biolabs | C2527I | |
| Recombinant DNA reagent | pET52b | EMD Millipore | 72554 | |
| Recombinant DNA reagent | pET52b/ His$_6$ Cas4 | *Lee et al., 2018* | N/A | |
| Recombinant DNA reagent | pSV272/ His$_6$-MBP-TEV Cas1 | *Lee et al., 2018* | N/A | |
| Recombinant DNA reagent | pSV272/ His$_6$-MBP-TEV Cas2 | *Lee et al., 2018* | N/A | |
| Recombinant DNA reagent | pRKSUF017 | *Takahashi and Tokumoto, 2002* | N/A | |
| Commercial assay or kit | QIAprep Spin Miniprep Kit | Qiagen | #27106 | |
| Commercial assay or kit | Wizard SV Gel and PCR Clean-up system | Promega | #A9282 | |
| Commercial assay or kit | Wizard Plus SV minipreps DNA purification system | Promega | #A1460 | |
| Commercial assay or kit | HisPur Ni-NTA Spin columns | Thermo Fisher Scientific | #88224 | |
| Commercial assay or kit | HisPur Ni-NTA resin | Thermo Fisher Scientific | #88223 | |
| Commercial assay or kit | HiTrap SP HP | GE Healthcare | #7115201 | |
| Commercial assay or kit | HiTrap Heparin HP | GE Healthcare | #17-0407-03 | |
| Commercial assay or kit | HiLoad 16/600 Superdex 200 | GE Healthcare | #28989335 | |
| Commercial assay or kit | HiLoad 16/600 Superdex 75 | GE Healthcare | #28989333 | |
| Commercial assay or kit | Sequenase Version 2.0 DNA Sequencing Kit | Thermo Fisher Scientific | 707701KT | |
| Software, algorithm | Scipion | *de la Rosa-Trevín et al., 2016* | scipion.i2pc.es | |
| Software, algorithm | Xmipp | *Abrishami et al., 2013*; *Sorzano et al., 2013*; *Vargas et al., 2013* | xmipp.i2pc.es | |
| Software, algorithm | RELION | *Scheres, 2012* | mrc-lmb.cam.ac.uk/reli | |
| Software, algorithm | RELION | *Scheres, 2012* | on/index.php/Main_Page | |

*Continued on next page*

*Continued*

| Reagent type (species) or resource | Designation | Source or reference | Identifiers | Additional information |
|---|---|---|---|---|
| Software, algorithm | Phyre2 | *Kelley et al., 2015* | sbg.bio.ic.ac.uk/phyre2/html/page.cgi?id = index | |
| Software, algorithm | Chimera | *Pettersen et al., 2004* | cgl.ucsf.edu/chimera/ | |
| Software, algorithm | Segger | *Pintilie et al., 2010* | cryoem.bcm.edu/cryoem/downloads/segger | |
| Other | Formvar/Carbon 400 mesh, Copper approx. grid hole size: 42 μm | Ted Pella, Inc. | 01754 F | |
| Other | *Pyrobaculum calidifontis* Cas4 | *Lemak et al., 2014* | PDB: 4R5Q | Deposited data |
| Other | *Archaeoglobus fulgidus* Cas1 | *Kim et al., 2013* | PDB: 4N06 | Deposited data |
| Other | *Bacillus halodurans* Cas2 | *Nam et al., 2012* | PDB: 4ES3 | Deposited data |
| Other | *Escherichia coli* Cas1-Cas2 | *Nuñez et al., 2014* | PDB: 4P6I | Deposited data |
| Other | *Escherichia coli* Cas1-Cas2-prespacer | *Nuñez et al., 2015b* | PDB: 5DS4 | Deposited data |
| Other | *Enterococcus faecalis* Cas1-Cas2-prespacer | *Xiao et al., 2017* | PDB: 5XVN | Deposited data |
| Other | *Enterococcus faecalis* Cas1-Cas2-full site | *Xiao et al., 2017* | PDB: 5XVP | Deposited data |
| Other | *Bacillus halodurans* Cas1-Cas2-target | This paper | EMDB-20127 | Deposited data |
| Other | *Bacillus halodurans* Cas4-Cas1-Cas2-target asymmetrical | This paper | EMDB-20128 | Deposited data |
| Other | *Bacillus halodurans* Cas4-Cas1-Cas2-target symmetrical | This paper | EMDB-20129 | Deposited data |
| Other | *Bacillus halodurans* Cas4-Cas1-Cas2-prespacer asymmetrical | This paper | EMDB-20130 | Deposited data |
| Other | *Bacillus halodurans* Cas4-Cas1-Cas2-prespacer symmetrical | This paper | EMDB-20131 | Deposited data |
| Chemical compound, drug | Agar | AMRESCO | #J637-1kg | |
| Chemical compound, drug | Carbenicillin disodium salt | RPI | #C46000-25.0 | |
| Chemical compound, drug | Kanamycin monosulfate | RPI | #K22000-25.0 | |
| Chemical compound, drug | Ampicillin | RPI | #A40040-100.0 | |

*Continued on next page*

*Continued*

| Reagent type (species) or resource | Designation | Source or reference | Identifiers | Additional information |
|---|---|---|---|---|
| Chemical compound, drug | Tetracycline HCl | RPI | #T17000-25.0 | |
| Chemical compound, drug | LB Broth (Miller) | Thermo Fisher Scientific | #BP1426-2 | |
| Chemical compound, drug | IPTG | RPI | #I56000-100.0 | |
| Chemical compound, drug | DTT | RPI | #D11000-100.0 | |
| Chemical compound, drug | Tryptone | RPI | #T60060-5000.0 | |
| Chemical compound, drug | Sodium chloride | AMRESCO | #7647–14.5 | |
| Chemical compound, drug | Yeast extract | Thermo Fisher Scientific | #BP1422-2 | |
| Chemical compound, drug | Agarose | Thermo Fisher Scientific | #BP160-500 | |
| Chemical compound, drug | HEPES | Thermo Fisher Scientific | #BP310-1 | |
| Chemical compound, drug | Sodium phosphate dibasic heptahydrate | Thermo Fisher Scientific | #S373-3 | |
| Chemical compound, drug | Glycerol | VWR analytical BDH | #BDH1172-4LP | |
| Chemical compound, drug | Imidazole | Thermo Fisher Scientific | #O31960599 | |
| Chemical compound, drug | PMSF | RPI | #P20270-25.0 | |
| Chemical compound, drug | Ferrous sulfate | Thermo Fisher Scientific | #I146-500 | |
| Chemical compound, drug | Ferric sulfate | Sigma | #F3388-250G | |
| Chemical compound, drug | L-Cysteine free base | MP Biomedicals | #194646 | |
| Chemical compound, drug | Manganese chloride tetrahydrate | Thermo Fisher Scientific | #M87-100 | |
| Chemical compound, drug | Potassium chloride | RPI | #D41000-2500.0 | |
| Chemical compound, drug | Brilliant blue R-250 | RPI | #B43000-50.0 | |

*Continued*

| Reagent type (species) or resource | Designation | Source or reference | Identifiers | Additional information |
|---|---|---|---|---|
| Chemical compound, drug | 40% Acrylamide/ Bis solution, 19:1 | Thermo Fisher Scientific | #BP1406-1 | |
| Chemical compound, drug | Urea | RPI | #U20200-25000.0 | |
| Chemical compound, drug | Boric acid | RPI | #B32050-5000.0 | |
| Chemical compound, drug | Tris | RPI | #T60040-5000.0 | |
| Chemical compound, drug | EDTA | Thermo Fisher Scientific | #BP120-1 | |
| Chemical compound, drug | 2X RNA loading dye | New England Biolabs | #B0363A | |
| Chemical compound, drug | T4 Polynucleotide Kinase | New England Biolabs | #M0201L | |
| Chemical compound, drug | [γ-$^{32}$P]-ATP | Perkin Elmer | #BLU502A250UC | |
| Chemical compound, drug | Phenol:Chloroform: Isoamyl Alcohol (25:24:1) | Thermo Fisher Scientific | #15593049 | |

## Protein purification

Cas1, Cas2, and Cas4 were purified as previously described (*Lee et al., 2018*). For complex formation using CRISPR hairpin DNA target, Cas1, Cas2, and hairpin DNA substrates or Cas4, Cas1, Cas2, and hairpin DNA substrates were mixed in equal molar ratios (1:1:1 or 1:1:1:1) and dialyzed against 1 L of buffer A (20 mM HEPES (pH 7.5), 100 mM NaCl, 5% glycerol, 2 mM DTT, and 2 mM MnCl$_2$) overnight at 4°C. The samples were loaded on a 5 mL HiTrap Q column (GE Healthcare) equilibrated with buffer A and the free Cas2 and free DNA were separated from the complex using a gradient of buffer B (20 mM HEPES (pH7.5), 1M NaCl, 5% glycerol, 2 mM DTT, 2 mM MnCl$_2$). Fractions containing all two or three proteins were pooled, concentrated, and further purified using a Superdex 75 10/30 GL column (GE Healthcare) in size exclusion buffer A (20 mM HEPES (pH 7.5), 100 mM KCl, 5% glycerol, 2 mM DTT, and 2 mM MnCl$_2$) for Cas1-Cas2-target complex and size exclusion buffer B (20 mM HEPES (pH 7.5), 250 mM KCl, 5% glycerol, 2 mM DTT, and 2 mM MnCl$_2$) for Cas4-Cas1-Cas2-target complex.

For complex formation using prespacer DNA substrate, 9.8 µM Cas4, 6.5 µM Cas1, 3.3 µM Cas2, and 5 µM prespacer were mixed (final approximate ratio of 3:2:1:1.5) in a final volume of 500 µl in size exclusion buffer B and incubated for 30 min at 4°C. The Cas4-Cas1-Cas2-prespacer complex was purified using a Superdex 75 10/30 GL column (GE Healthcare) in size exclusion buffer B.

## DNA substrate preparation

All oligonucleotides were synthesized by Integrated DNA Technologies or Thermo Scientific. Sequences of DNA substrates are shown in *Table 1*. All DNA substrates were purified on 10% urea-PAGE. Double-stranded DNA was hybridized by heating to 95°C for 5 min followed by slow cooling to room temperature in oligo annealing buffer (20 mM HEPES (pH 7.5), 25 mM KCl, 10 mM MgCl$_2$). Prespacers were labeled with [γ-$^{32}$P]-ATP (PerkinElmer) and T4 polynucleotide kinase (NEB) for 5'-end labelling. Excess ATP was removed using Illustra Microspin G-25 columns (GE Healthcare).

**Table 1.** Oligonucleotides used in this study.

| Sequence (5' → 3') | Description |
|---|---|
| *GATTTTCGCT*GTCGCACTCTTCATGGGTGCGTGGATTGAAAT ATTGAcgatagTCAAT**ATTTCAATCCACGCACCCATGAAGAGTGC GAC***AGCGAAAATC* | CRISPR hairpin target[*] |
| *GATTTTCGCT*GTCGCACTCTTCATGGGTGCGTGGATTGAAAT ATTGAGGTAGGTATTG | Mini-CRISPR array |
| CAATACCTACCTCAATATTTCAATCCACGCACCC ATGAAGAGTGCGACAGCGAAAATC | RC[†] |
| CGTAGCTGAGGACCACCAGAACAG TTTT<u>GAA</u>TTTTTTTT | 15-nt 3' overhang prespacer, 4-nt between duplex and PAM[††] |
| CGTAGCTGAGGACCACCAGAACAG TT<u>GAA</u>TTTTTTTTTT | 2-nt between duplex and PAM |
| CGTAGCTGAGGACCACCAGAACAG TTTTTT<u>GAA</u>TTTTTT | 6-nt between duplex and PAM |
| CGTAGCTGAGGACCACCAGAACAG TTTTTTTT<u>GAA</u>TTTT | 8-nt between duplex and PAM |
| CTGTTCTGGTGGTCCTCAGCTACG TTTT<u>GAA</u>TTTTTTTT | RC of previous four oligos |
| *GATTTTCGCT*GTCGCACTCTTCATGGGTGCGTGGATTGAAATATTGA | CRISPR DNA substrate |
| TCAATATTTCAATCCACGCACCCATGAAGAGTGCGACAGCGAAAATC | RC |
| GCGTAGCTGAGGACCACCAGAACAGTTTT<u>GAA</u>TTTTTTTTTTTTTTTTTTTT | 25-nt 3' overhang prespacer |
| GCGTAGCTGAGGACCACCAGAACAG | 25 bp duplex |
| CTGTTCTGGTGGTCCTCAGCTACGC | RC |
| GCGTAGCTGAGGACCTTTTTTTTTTTTT<u>GAA</u>TTTTTTTTTTTTTTTCAGGT CGACAAGCTTG | T-rich ssDNA prespacer |
| CAAGCTTGTCGACCTGAAAAAAAAAAAAAAATTCAAAAAAAAAAAAA GGTCCTCAGCTACGC | RC |
| CTAGTATGATCATGTCCAACGAATCAATACCTACCTCAATGAACGGAT | 48 bp duplex |
| ATCCGTTCATTGAGGTAGGTATTGATTCGTTGGACATGATCATACTAG | RC |
| GCGTAGCTGAGGACCTTTTTTTTTTTTTTTTTTT<u>GAA</u>TT<u>GAA</u>TT<u>GAA</u> TTTTTTTTTTTTTTTTTTTGACAAGCTTGCGACA | 3 PAM sites interspersed in 2-nt |
| TGTCGCAAGCTTGTCAAAAAAAAAAAAAAAAAAATTCAATTCAATTCA AAAAAAAAAAAAAAAAAAGGTCCTCAGCTACGC | RC |
| GCGTAGCTGAGGACCTTTTTTTTTTTTTTTTTTTTT<u>GAAGAAGAA</u>TTTTT TTTTTTTTTTTTTGACAAGCTTGCGACA | 3 PAM sites without spacing |
| TGTCGCAAGCTTGTCAAAAAAAAAAAAAAAAAAAAATT<u>CTTCTT</u>CAAA AAAAAAAAAAAAAAAGGTCCTCAGCTACGC | RC |
| GCGTAGCTGAGGACCTTTTTTTTTT<u>GAA</u>TTTTTTTTTT<u>GAA</u>TTTT TTTTTT<u>GAA</u>TTTTTTTTTTGACAAGCTTGCGACA | 3 PAM sites interspersed with 10-nt |
| TGTCGCAAGCTTGTCAAAAAAAAAATTCAAAAAAAAAATTCAAAAA AAAAATTCAAAAAAAAAAGGTCCTCAGCTACGC | RC |
| GCGTAGCTGAGGACCTTTTTTTTTTTT<u>GAA</u>TTTTTTTT<u>GAA</u>TTTTTTTT <u>GAA</u>TTTTTTTTTTTTGACAAGCTTGCGACA | 3 PAM sites interspersed with 8-nt |
| TGTCGCAAGCTTGTCAAAAAAAAAAAATTCAAAAAAAATTCAAAAAAAA TTCAAAAAAAAAAAAGGTCCTCAGCTACGC | RC |
| GCGTAGCTGAGGACCTTTTTTTTTTTTTT<u>GAA</u>TTTTTT<u>GAA</u>TTTT TT<u>GAA</u>TTTTTTTTTTTTTTGACAAGCTTGCGACA | 3 PAM sites interspersed with 6-nt |
| TGTCGCAAGCTTGTCAAAAAAAAAAAAAAATTCAAAAAATTCAAAA AATTCAAAAAAAAAAAAAAGGTCCTCAGCTACGC | RC |
| GCGTAGCTGAGGACCTTTTTTTTTTTTTTTT<u>GAA</u>TTTT<u>GAA</u>TTTT<u>GAA</u> TTTTTTTTTTTTTTTTGACAAGCTTGCGACA | 3 PAM sites interspersed with 4-nt |
| TGTCGCAAGCTTGTCAAAAAAAAAAAAAAAAATTCAAAATTCAAAA TTCAAAAAAAAAAAAAAAAGGTCCTCAGCTACGC | RC |
| GCGTAGCTGAGGACCTATATATATATAT<u>GAA</u>TATATATATATATA CAGGTCGACAAGCTTG | AT-rich ssDNA prespacer |
| CAAGCTTGTCGACCTGTATATATATATATATTCATATATATAT ATAGGTCCTCAGCTACGC | RC |
| GCGTAGCTGAGGACCTTGGTATTCAACA<u>GAA</u>TTTTTTTTTTTTTTCA GGTCGACAAGCTTG | Non-T-rich upstream/T rich downstream ssDNA prespacer |

*Table 1 continued on next page*

*Table 1 continued*

| Sequence (5′ → 3′) | Description |
|---|---|
| CAAGCTTGTCGACCTGAAAAAAAAAAAAAAATTCTGTTGAATACCAAG GTCCTCAGCTACGC | RC |
| GCGTAGCTGAGGACCTTTTTTTTTTTTTTTGAACTCGTATTCAACAG CAGGTCGACAAGCTTG | T-rich upstream/non T-rich downstream ssDNA prespacer |
| CAAGCTTGTCGACCTGCTGTTGAATACGAGTTCAAAAAAAAAAAAAA GGTCCTCAGCTACGC | RC |
| GCGTAGCTGAGGACCTTGGTATTCAACAGAACTCGTATTC AACAGCAGGTCGACAAGCTTG | Non-T-rich up- and downstream ssDNA prespacer |
| CAAGCTTGTCGACCTGCTGTTGAATACGAGTTCTGTTGAATACCAA GGTCCTCAGCTACGC | RC |
| GCGTAGCTGAGGACC | Primer used for ddNTP Sanger sequencing |
| GCGTAGCTGAGGACCCGTGGCACCGACATGGCATTTTTNNNNGAA TTTTTGCTGGGCGCTAAGGGACAACTCCAGGTCGACAAGCTTG | NNNN on upstream region |
| GCGTAGCTGAGGACCCGTGGCACCGACATGGCAGTTTTTNNNGAA TTTTTGCTGGGCGCTAAGGGACAACTCCAGGTCGACAAGCTTG | NNN on upstream region |
| GCGTAGCTGAGGACCCGTGGCACCGACATGGCAGGTTTTTNNGAA TTTTTGCTGGGCGCTAAGGGACAACTCCAGGTCGACAAGCTTG | NN on upstream region |
| GCGTAGCTGAGGACCCGTGGCACCGACATGGCAGGCTTTTTNGAA TTTTTGCTGGGCGCTAAGGGACAACTCCAGGTCGACAAGCTTG | N on upstream region |
| GCGTAGCTGAGGACCCGTGGCACCGACATGGCATTTTTGAANNNN TTTTTGCTGGGCGCTAAGGGACAACTCCAGGTCGACAAGCTTG | NNNN on downstream region |
| GCGTAGCTGAGGACCCGTGGCACCGACATGGCATTTTTGAANNNTTTTT CGCTGGGCGCTAAGGGACAACTCCAGGTCGACAAGCTTG | NNN on downstream region |
| GCGTAGCTGAGGACCCGTGGCACCGACATGGCATTTTTGAANN TTTTTCAGCTGGGCGCTAAGGGACAACTCCAGGTCGACAAGCTTG | NN on downstream region |
| GCGTAGCTGAGGACCCGTGGCACCGACATGGCATTTTTGAAN TTTTTCATGCTGGGCGCTAAGGGACAACTCCAGGTCGACAAGCTTG | N on downstream region |
| CAAGCTTGTCGACCTG | Primer used for primer extension |
| TCGTCGGCAGCGTCAGATGTGTATAAGAGACAGCAAGCTTGTCGACCTG | Primer used for amplification-Forward |
| GTCTCGTGGGCTCGGAGATGTGTATAAGAGACAGGCGTAGCTGAGGACC | Primer used for amplification-Reverse |

[*]For CRISPR oligonucleotides, leader is in italics, repeat is in bold, and spacer is in plain uppercase font. For hairpin, the loop region is in lowercase.

[†]RC = reverse complement of previous oligonucleotide.

[††]For cleavage substrates, PAM sequences are underlined.

DOI: https://doi.org/10.7554/eLife.44248.020

## Negative stain EM sample preparation and data collection

To prepare grids for EM imaging, Cas1-Cas2-target, Cas4-Cas1-Cas2-target or Cas4-Cas1-Cas2-prespacer were diluted to ~100 nM, and 3 µL of sample was applied to a glow-discharged copper 400-mesh continuous carbon grid for one minute at room temperature. The excess sample was blotted with Whatman filter paper, followed by immediate application of 3 µL 2% (w/v) uranyl formate. The excess stain was blotted, followed by immediate application of 3 µL 2% uranyl formate. This step was repeated once more. The grids were allowed to dry for at least 5 min prior to imaging.

Images were collected on a 200 keV JEOL 2100 transmission electron microscope equipped with a Gatan OneView camera at a nominal magnification of 60,000x and pixel size of 1.9 Å. The electron dose was between 30 and 40 electrons/Å$^2$. For each sample, images (200 for Cas1-Cas2-target and Cas4-Cas1-Cas2-target and 93 for Cas4-Cas1-Cas2-prespacer) were collected manually at a defocus range of 1–2.5 µm.

## Image processing and single-particle analysis

All image processing and analysis was performed in Scipion v. 1.2 (*de la Rosa-Trevín et al., 2016*) (RRID:SCR_016738) (available at http://scipion.i2pc.es/). The contrast transfer function (CTF) for each micrograph was estimated using CTFFIND4 (*Rohou and Grigorieff, 2015*) (RRID:SCR_016732). For

each sample,~200 particles were picked using Xmipp manual picking, followed by automated picking using the manually picked particles as a training set (*Abrishami et al., 2013*; *Sorzano et al., 2013*; *Vargas et al., 2013*). In total, 95,669 Cas1-Cas2-target, 115,445 Cas4-Cas1-Cas2-target and 32,494 Cas4-Cas1-Cas2-prespacer particles were present in the initial data set. Particles were extracted using a 160 × 160 pixel box. To reduce computational requirements, the particles were down sampled by a factor of 2 to a final box size of 80 × 80 pixels (~152×152 Å). The particles were normalized and subjected to reference-free 2D classification using RELION 2.1 (*Scheres, 2012*) (RRID:SCR_016274) (*Figure 2—figure supplement 1B*). The initial 100 class averages were inspected, and averages with clear structural features, the largest number of particles and size consistent with the molecular weight of the complex were selected for further analysis. These particles were subjected to a second round of 2D classification into 50 classes using RELION to further clean the particles. After selection of the best 2D classes, the Cas1-Cas2-target dataset contained 34,626 particles, the Cas4-Cas1-Cas2-target dataset contained 49,173 particles, and the Cas4-Cas1-Cas2-prespacer dataset contained 19,620 particles.

Particles were next subjected to 3D classification in RELION using the X-ray crystal structure of *E. coli* Cas1-Cas2 (PDB: 4P6I (*Nuñez et al., 2014*)) low-pass-filtered to 90 Å as a starting model (*Figure 2—figure supplement 2*). The target-bound complex datasets were initially classified into six classes, while the prespacer-bound complex was classified into five classes due to the lower number of starting particles. For Cas1-Cas2-target, 11,636 particles from two similar 3D classes with the clearest density were combined (*Figure 2—figure supplement 2A*). These particles were subjected to 3D refinement using RELION, and the refined volume was used to create a 3D mask. The refined particles were subjected to a second round of classification into three classes using the 3D mask as a reference mask (*Figure 2—figure supplement 2A*). A class containing 5279 particles with the clearest density was selected. This class contained clear C2 symmetry. These particles were subjected to 3D refinement while enforcing C2 symmetry and using the 3D mask yielding the final reconstruction.

For Cas4-Cas1-Cas2-target, 37,051 particles from four out of six initial 3D classes that appeared to contain Cas1-Cas2 with additional density were selected for further refinement (*Figure 2—figure supplement 2B*). These particles were subjected to 3D refinement and then further classified into four 3D classes. The resulting classes had clearly defined extra density in comparison to the Cas1-Cas2 3D reconstruction. For two of these classes, the density displayed clear C2 symmetry (*Figure 2—figure supplement 2B*), with extra density extending from each Cas1 lobe, while for the other two classes, the extra density was only observed extending from one Cas1 lobe. Particles (18,290) from the two symmetrical 3D classes were combined and subjected to 3D refinement while enforcing C2 symmetry. Particles (9,160) from one of the asymmetrical 3D classes were subjected to 3D refinement with C1 symmetry.

For Cas4-Cas1-Cas2-prespacer, 11,001 particles from two out of five initial 3D classes that appeared to contain density in addition to the Cas1-Cas2 core were selected for further refinement (*Figure 2—figure supplement 2C*). These particles were subjected to 3D refinement and then further classified into three 3D classes. The resulting classes had varying degrees of extra density. Class 1 (3,682 particles) contained no apparent extra density in comparison to Cas1-Cas2-target, while the other two classes resembled the asymmetrical (4668 particles) and symmetrical (2651 particles) Cas4-Cas1-Cas2-target densities. Particles from the symmetrical 3D class were subjected to 3D refinement while enforcing C2 symmetry. Particles from the asymmetrical 3D classes were subjected to 3D refinement with C1 symmetry. For all Cas4-Cas1-Cas2 samples, the refined 3D reconstructions were used to create 3D masks, and each set of particles was subjected to a final round of refinement using the 3D mask as reference mask.

The resolutions of the final 3D reconstruction were 22.1 Å, 19.7 Å, 21.4 Å, 21.6 Å and 24.4 Å for Cas1-Cas2-target, symmetrical Cas4-Cas1-Cas2-target, asymmetrical Cas4-Cas1-Cas2-target, symmetrical Cas4-Cas1-Cas2-prespacer and asymmetrical Cas4-Cas1-Cas2-prespacer, respectively, based on Fourier Shell Correlation (FSC) at a cutoff of 0.5 (*Figure 2—figure supplement 1C–D*). The 0.5 FSC criterion was used to ensure that resolution was not overestimated. The Euler angle plots for the final 3D reconstructions revealed some preferred orientations but indicated a wide angular distribution in the data (*Figure 2—figure supplement 1E*).

## Structural modelling

Volumes were segmented using Segger (*Pintilie et al., 2010*) in UCSF Chimera (*Pettersen et al., 2004*). Structural models for the *B. halodurans* Cas1 and Cas4 sequences were predicted using the Phyre2 webserver (http://www.sbg.bio.ic.ac.uk/phyre2/html/page.cgi?id=index) (*Kelley et al., 2015*). The top results provided structural models based on the closest homologs of BhCas1 and BhCas4 available in the protein databank. For Cas1, the closest homolog is Cas1 from *Archaeoglobus fulgidus* (PDB 4N06, 28% identical, 65% similar) (*Kim et al., 2013*). For Cas4, the closest available homolog structure is from *Pyrobaculum calidifontis* (PDB 4R5Q, 15% identical, 44% similar) (*Lemak et al., 2014*). The crystal structure of *B. halodurans* Cas2 (PDB 4ES3) was used for fitting to density assigned to Cas2 (*Nam et al., 2012*). Fitting of individual copies of Cas1 dimers, Cas2 dimer or Cas4 monomers into assigned densities was performed using the 'Fit in Segments' tool in Segger within UCSF Chimera (RRID:SCR_004097) (*Pettersen et al., 2004*; *Pintilie et al., 2010*). For Cas4, the top four fits are shown for segmented volumes of the symmetrical and asymmetrical Cas4-Cas1-Cas2-target complexes in *Figure 2—figure supplement 4D–E*. The cross-correlation score provided by Segger is reported in the figure. The distance between the alpha carbon atom of the Cas1 H234 and Cas4 K110 active site residues was measured using the 'Distances' tool in UCSF Chimera. For analyzing fit of Cas1-Cas2 crystal structures in the Cas1-Cas2-target density, the protein subunits of the X-ray crystal structure of *E. faecalis* Cas1-Cas2 bound to prespacer (PDB: 5XVN [*Xiao et al., 2017*]) or *E. coli* Cas1-Cas2 bound to prespacer (PDB 5DS4 [*Nuñez et al., 2015b*]) were docked into the final 3D reconstruction using the Fit to Segments tool in Segger (*Figure 2—figure supplement 3*).

## Prespacer processing assays

Prespacer processing assays were performed using 25 nM of 5'-radiolabeled substrate with 500 nM Cas4, 200 nM Cas1, 200 nM Cas2 and 1 µM mini-CRISPR DNA (as indicated) or indicated amount of activating dsDNA in integration buffer (20 mM HEPES (pH 7.5), 100 mM KCl, 5% glycerol, 2 mM DTT, and 2 mM MnCl$_2$). The proteins and activating dsDNA were incubated on ice for 10 min prior to addition of the 5'-labeled substrate. All reactions were performed at 65°C for 20 min. Reactions were quenched with 2X RNA loading dye (NEB) and heated at 95°C for 5 min followed by cooling on ice for 3 min. Samples were run on 12% urea-PAGE. The gels were dried and imaged using phosphor screens on a Typhoon imager.

Sequenase Version 2.0 DNA sequencing kit (Applied Biosystems) was used for Sanger sequencing lanes. Samples were prepared by hybridizing template with 5'-radiolabeled primer at 65°C for 2 min and slowly cooling to RT for 30 min. Samples were incubated with the chain terminators (ddGTP, ddCTP, ddATP or ddTTP) at 37°C for 15 min and quenched with 2X RNA dye. The cleaved products were prepared in the presence of 1.2 µM Cas4, 600 nM Cas1, 600 nM Cas2, and 1 µM of activator 48 bp dsDNA and quenched with 2X RNA loading dye. All samples were heated at 95°C for 5 min followed by cooling on ice for 3 min. Samples were run on 0.4 mm 8% urea-PAGE. The gels were dried and imaged using a Typhoon imager.

Cas4 cleavage assays for the panel of duplex substrates in which the PAM position was varied within the single-strand overhang were performed as described above, with the following alterations. For preparing duplex substrates, 25 nM 5' radiolabeled top strand in which the PAM position was varied was annealed to 50 nM unlabeled bottom strand in buffer (20 mM HEPES (pH 7.5), 100 mM KCl, 5% glycerol, 2 mM DTT, 2 mM MnCl2,). The hybridization reactions were incubated at 95°C for 3 min followed by slow cooling to room temperature. A final concentration of 5 nM radiolabeled substrate was used in the reaction. Processing reactions were performed in triplicate, and the intensity of bands was measured by densitometry using ImageJ (*Schneider et al., 2012*). The fraction cleaved was calculated by dividing the product band by the sum of both bands. The values from three replicates were averaged, and error is reported as standard deviation between the replicates.

## Analysis of PAM-flanking sequence depletion

The ssDNA substrates containing between 1–4 degenerate nucleotides upstream or downstream or the PAM were purified with 10% urea-PAGE. For cleavage, 20 nM of each substrate was incubated with 600 nM Cas1, 600 nM Cas2 and 1 µM activator dsDNA in the presence or absence of 1.2 µM Cas4. Samples without Cas4 were used as negative control. Three separate samples were prepared

for each condition and treated as separate replicates. After ssDNA cleavage, the uncleaved products were hybridized with a primer that was complementary to the 3' end and extended using Klenow Fragment (NEB). Samples were extracted with phenol-chloroform-isoamyl alcohol and precipitated with ethanol. The samples were amplified by PCR using Platinum SuperFi DNA polymerase (Thermo) with primers containing Nextera adapters followed by a second round of PCR with primers containing i5 and i7 barcodes. Amplification products were analyzed on 2% SYBR Safe stained agarose gels and quantified using densitometry. Samples were mixed in equal quantities and were run on 2% agarose gel. The band was excised and DNA was purified using QIAquick Gel Extraction kit (Qiagen). Samples were submitted for Illumina MiSeq sequencing to the Iowa State University DNA facility.

The de-multiplexed datasets were analyzed separately to determine the relative read counts for each possible sequence in the degenerate regions. The degenerate regions of the sequences were cut from the reads, and the number of counts for each unique sequence was determined using bash commands. The reads were normalized by dividing the number of reads for each sequence by the total number of reads for the dataset. The normalized reads from three replicates were averaged for each substrate for reactions performed in the absence or presence of Cas4, and error was calculated as standard deviation between the replicates.

## Acknowledgements

We thank Sashital lab members for helpful discussions; David Taylor for helpful comments on the manuscript; Levi Baber and members of the ResearchIT group at Iowa State University for computational support; Tracey Stewart and the Roy J Carver High Resolution Microscopy Facility for technical assistance with negative stain EM data collection; and members of the Walter Moss lab at Iowa State University for sharing sequencing gel electrophoresis equipment. This work was supported by an NIH R01 grant (GM115874) to DGS.

## Additional information

### Funding

| Funder | Grant reference number | Author |
| --- | --- | --- |
| National Institute of General Medical Sciences | GM115874 | Dipali G Sashital |

The funders had no role in study design, data collection and interpretation, or the decision to submit the work for publication.

### Author contributions

Hayun Lee, Conceptualization, Formal analysis, Investigation, Methodology, Writing—original draft, Writing—review and editing, Performed complex formation of Cas4-Cas1-Cas2-target, prespacer and ssDNA cleavage experiments, and ssDNA specificity assays; Yukti Dhingra, Formal analysis, Investigation, Methodology, Writing—review and editing, Performed Cas4-Cas1-Cas2-prespacer complex formation, negative stain EM data collection, and processing assay for duplex substrates with varied PAM positions; Dipali G Sashital, Conceptualization, Formal analysis, Funding acquisition, Investigation, Methodology, Writing—original draft, Writing—review and editing, Performed negative stain EM data collection and analysis and high-throughput sequencing analysis

### Author ORCIDs

Hayun Lee http://orcid.org/0000-0002-3365-8955
Dipali G Sashital http://orcid.org/0000-0001-7681-6987

### Decision letter and Author response

Decision letter https://doi.org/10.7554/eLife.44248.047
Author response https://doi.org/10.7554/eLife.44248.048

# Additional files

## Supplementary files
• Transparent reporting form
DOI: https://doi.org/10.7554/eLife.44248.021

## Data availability
The negative-stain EM volumes for the Cas1-Cas2-target, asymmetrical Cas4-Cas1-Cas2-target, symmetrical Cas4-Cas1-Cas2-target, asymmetrical Cas4-Cas1-Cas2-prespacer and symmetrical Cas4-Cas1-Cas2-prespacer complexes have been deposited to EMDB under the accession numbers EMDB-20127, EMDB-20128, EMDB-20129, EMDB-20130 and EMDB-20131, respectively.

The following datasets were generated:

| Author(s) | Year | Dataset title | Dataset URL | Database and Identifier |
|---|---|---|---|---|
| Lee H, Dhingra Y, Sashital DG | 2019 | Bacillus halodurans Cas1-Cas2-target | http://www.ebi.ac.uk/pdbe/entry/emdb/EMD-20127 | Electron Microscopy Data Bank, EMD-20127 |
| Lee H, Dhingra Y, Sashital DG | 2019 | Bacillus halodurans Cas4-Cas1-Cas2-target asymmetrical | http://www.ebi.ac.uk/pdbe/entry/emdb/EMD-20128 | Electron Microscopy Data Bank, EMD-20128 |
| Lee H, Dhingra Y, Sashital DG | 2019 | Bacillus halodurans Cas4-Cas1-Cas2-target symmetrical | http://www.ebi.ac.uk/pdbe/entry/emdb/EMD-20129 | Electron Microscopy Data Bank, EMD-20129 |
| Lee H, Dhingra Y, Sashital DG | 2019 | Bacillus halodurans Cas4-Cas1-Cas2-prespacer asymmetrical | http://www.ebi.ac.uk/pdbe/entry/emdb/EMD-20130 | Electron Microscopy Data Bank, EMD-20130 |
| Lee H, Dhingra Y, Sashital DG | 2019 | Bacillus halodurans Cas4-Cas1-Cas2-prespacer symmetrical | http://www.ebi.ac.uk/pdbe/entry/emdb/EMD-20131 | Electron Microscopy Data Bank, EMD-20131 |

The following previously published datasets were used:

| Author(s) | Year | Dataset title | Dataset URL | Database and Identifier |
|---|---|---|---|---|
| Nocek B, Skarina T, Lemak S, Brown G, Savchenko A, Joachimiak A, Yakunin A, Midwest Center for Structural Genomics (MCSG) | 2014 | Pyrobaculum calidifontis Cas4 | https://www.rcsb.org/structure/4R5Q | Protein Data Bank, 4R5Q |
| Kim TY, Shin M, Yen LHT, Kim JS | 2013 | Crystal structure of Cas1 from Archaeoglobus fulgidus and its nucleolytic activity | https://www.rcsb.org/structure/4N06 | Protein Data Bank, 4N06 |
| Ke A, Nam KH | 2012 | Double-stranded Endonuclease Activity in B. halodurans Clustered Regularly Interspaced Short Palindromic Repeats (CRISPR)-associated Cas2 Protein | https://www.rcsb.org/structure/4ES3 | Protein Data Bank, 4ES3 |
| Nunez JK, Kranzusch PJ | 2014 | Crystal structure of the Cas1-Cas2 complex from Escherichia coli | https://www.rcsb.org/structure/4P6I | Protein Data Bank, 4P6I |
| Nunez JK, Harrington LB, Kranzusch PJ, Engelman AN, Doudna JA | 2015 | Crystal structure the Escherichia coli Cas1-Cas2 complex bound to protospacer DNA | https://www.rcsb.org/structure/5DS4 | Protein Data Bank, 5DS4 |
| Xiao Y, Ng S, Nam KH | 2017 | E. far Cas1-Cas2/prespacer binary complex | https://www.rcsb.org/structure/5XVN | Protein Data Bank, 5XVN |
| Xiao Y, Ng S, Nam KH | 2017 | E. fae Cas1-Cas2/prespacer/target ternary complex revealing the fully integrated states | https://www.rcsb.org/structure/5XVP | Protein Data Bank, 5XVP |

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
