## [Decision Letter]

Thank you for submitting your article "The Cas4-Cas1-Cas2 complex mediates precise prespacer processing during CRISPR adaptation" for consideration by *eLife*. Your article has been reviewed by three peer reviewers, one of whom is a member of our Board of Reviewing Editor, and the evaluation has been overseen by Cynthia Wolberger as the Senior Editor. The following individual involved in review of your submission has agreed to reveal their identity: Malcolm F White (Reviewer #2).

The reviewers have discussed the reviews with one another and the Reviewing Editor has drafted this decision to help you prepare a revised submission. While all reviewers recognize that the significance of the biochemical work, the consensus is that the conclusions from the EM work need to be strengthened, either by improving the structure resolution, or through extensive mutagenesis validation. A summary of their review comments can be found below.

Summary:

Cas4 is prevalent in many CRISPR systems. The Sashital group and several other labs defined its important function in processing the 3ʹ-end of prespacer in a PAM-dependent and Cas1-Cas2-dependent fashion. The authors previously reported the low-resolution EM structure of the Cas4-Cas1 complex. Important questions remain to be answered. Does PAM-dependent prespacer processing take place in this complex? How does the processing lead to directional spacer integration into the CRISPR array? In this manuscript, Lee et al. reconstituted the Cas4-Cas1-Cas2 complex by including a dsDNA substrate, and presented a 22 Å negative-staining EM structure of the *Bacillus halodurans* Cas4-Cas1-Cas2 structure. Cas4 was found to bind Cas1 in either symmetrical or asymmetrical fashion, suggesting that its interaction with Cas1 is dynamic and could be influenced by the presence of a PAM sequence in the ssDNA overhang. The proximity of the nuclease center in Cas4 to the integration active site of Cas1 suggests a mechanism for substrate handoff following PAM-dependent processing. The PAM-dependent prespacer processing more active and precise in the context of this complex,. The authors further propose a mechanism explaining how Cas4-mediated PAM-end processing may lead to directional integration of the mature prespacer into the CRISPR locus.

Essential revisions:

Major points on the negative staining EM work:

1) The current resolution of the EM structure leaves room for ambiguous interpretations. Reviewers feel that conclusions of this work would be significantly strengthened by improving the resolution to a range (i.e. 6-10 Å) that allows unambiguous docking of high-resolution structures. If it turns out improving the structure resolution is not practical at this point, the authors are strongly encouraged to carry out a set of mutagenesis experiments to support their docking model. Given that two different sets of Cas4-Cas1 contacts were observed in the Cas4-Cas1 and Cas4-Cas1-Cas2 structures, the authors may want to mutagenize the interface residues in each of the structure, and evaluate the functional consequences.

2) The Figures 2 and Figure 2—figure supplement 4 were not clear and the reviewers could not get a good handle on the orientation of the Cas4 proteins relative to the DNA. Figure 2 shows 20 Å EM reconstruction of the Cas1-2 and Cas1-2-4 complexes. At this resolution, the Cas1-2 complex model is probably OK and matches known structures. The extra "blob" in the Cas1-2-4 complexes is assigned to Cas4, however it is smaller than the smallest known Cas4, the monomeric protein from *P. calidifontis*. It also does not fit with the EM structure of the Cas1-4 complex previously reported by the same authors (Lee, 2018). Although the FeS cluster appeared to help, this assignment should be treated with caution. In particular, the orientation chosen for the modelled Cas4 looks close to arbitrary and should not be used as a basis for further interpretation about active site positioning.

3) The absence of density corresponding to the CRISPR DNA is troublesome and needs to be discussed by the authors. What is the rationale of using the CRISPR leader-repeat-spacer instead of the prespacer duplex to promote the formation of the Cas4-Cas1-Cas2 complex? It appears that initially the authors reasoned that target-binding may promote complex formation, but their later data seems to argue that the Cas4-Cas1-Cas2 complex contacted the DNA rather nonspecifically, and through the prespacer-binding interface. If this is the case, to avoid possible misconceptions, the authors should make it clear that the contact does not reflect how Cas1-Cas2 recognizes the target sequence.

Major points on the biochemistry work:

1) The biochemistry experiments in this manuscript were done using prespacer substrates containing a 'perfect' duplex with ss-overhangs provided either in cis or in trans. Such substrates are rarely found in vivo. In cases when the PAM is buried in the dsDNA region, can Cas4 process the dsDNA and correctly cleave after the 3'-PAM? Can the product be integrated into the CRISPR substrate?

2) What would happen when a ss-PAM is 1-2 nucleotide away (too far or too close) from the 5-nt spacing? Does the Cas4 cleavage dictate where the integration happens?

3) Figure 3 presents some nice data showing that prespacers are cleaved by Cas4 and that this is stimulated by the presence of ds CRISPR DNA. The data on ssDNA substrates, which is discussed in depth, is likely not that relevant to the adaptation process and activity is much weaker for these substrates. In the figure legend, part C is not described. This figure does lack an important control – use of a duplex substrate with ss overhangs that lacks a PAM sequence to discriminate between a ruler mechanism and a sequence specific mechanism for cleavage by Cas4 of the relevant prespacer structures.

4) Figure 4 and 5 show convincing data supporting PAM-dependent cleavage activity by Cas1-2-4 on ssDNA substrates. This provides important new information on PAM removal by Cas4 during the capture process. It is unclear why this work has not been done with the relevant substrate, the gapped duplex shown in Figure 3B, which is cleaved more efficiently that non-relevant ssDNA. It would considerably strengthen the paper if the authors could provide some evidence that the same PAM-specific cleavage is observed with relevant duplex substrates, and this should not be hard to do.

---

## [Author Response]

Essential revisions:Major points on the negative staining EM work:1) The current resolution of the EM structure leaves room for ambiguous interpretations. Reviewers feel that conclusions of this work would be significantly strengthened by improving the resolution to a range (i.e. 6-10 Å) that allows unambiguous docking of high-resolution structures. If it turns out improving the structure resolution is not practical at this point, the authors are strongly encouraged to carry out a set of mutagenesis experiments to support their docking model. Given that two different sets of Cas4-Cas1 contacts were observed in the Cas4-Cas1 and Cas4-Cas1-Cas2 structures, the authors may want to mutagenize the interface residues in each of the structure, and evaluate the functional consequences.

The reviewers’ concern that the structural model is ambiguous is valid; however, the suggested improvement to resolution could not be completed in the timescale of revisions (see comment 1 below) and the mutagenesis experiments are extremely challenging due to lack of high-resolution structures for *B. halodurans* Cas1 and Cas4 (see comment 2 below). Because we were unable to complete these experiments requested by the reviewers, we have made extensive changes to our structural modelling experiments, which are outlined in our response to concern #2.

1) An improvement in resolution would certainly be beneficial but will require considerable investment of time, as well as resources that are not readily available to us. The resolution range suggested by the reviewers is outside the theoretical limits of negative stain EM and would therefore require cryo-EM. This instrumentation is not currently available to our lab, which is why we pursued negative stain EM. We have also initiated an off-site collaboration in the hopes of obtaining a higher-resolution structure by cryo-EM. Unfortunately, as is often the case when moving from negative stain to cryo, it seems that cryo-EM structure determination from these samples will require considerable sample optimization. Because we are pursuing this work through collaboration, it is not possible to complete this optimization in the timeframe allowed for manuscript revisions. We also want to note that a high-resolution structure of BhCas4 is not available (see next comment) and Cas4 is a compact protein with few distinctive structural features. So, it is unclear whether the resolution range suggested by the reviewers would enable unambiguous docking of the Pcal Cas4 structure in the BhCas4 density. Generally, < 6 Å resolution is required to clearly define all secondary-structural features, especially beta-strands, in a protein. We believe that this goal will have to remain the subject of future studies.

2) We appreciate the reviewers’ suggestion of testing mutations in the predicted Cas4-Cas1 interface. Indeed, prior to our initial submission of this manuscript, we had attempted to predict Cas4 and Cas1 residues that could disrupt interactions upon mutation based on our structural models. However, it is important to note that high-resolution structures are not available for the *B. halodurans* Cas1 and Cas4 sequences. While a structure of a relatively close Cas1 homolog (*Archaeoglobus fulgidus*, PDB 4N06, 28% identical, 65% similar) is available, the closest available Cas4 homolog structure (*Pyrobaculum calidifontis*, PDB 4R5Q, 15% identical, 44% similar) has several predicted structural divergences (Figure 2 —figure supplement 4A and B).

Most importantly, the entire N-terminal helix of PcCas4 is likely not present in BhCas4, while a ~20 amino acid C-terminal domain in BhCas4 is not present in the PcCas4 structure (Figure 2—figure supplement 4A). In an attempt to resolve these structural differences, we used the Phyre2 server to predict structural models of BhCas1 and BhCas4 (Figure 2—figure supplement 4B). However, we are not confident in the exact placement of amino acid sidechains within these models, especially given the lack of a potential BhCas4 C-terminal domain in the structural model. Cas4-Cas1 disruption experiments would test for negative results (i.e. loss of complex formation or Cas4 processing) and the lack of a high-resolution BhCas4 structure to guide putatively disruptive mutations could lead to false-positive loss of function (e.g. due to protein misfolding) or false negatives (e.g. due to compensatory structural rearrangements). Thus, mutagenesis experiments are premature at this point and could exacerbate the risk of overinterpreting the structural model. We have instead provided more information on potential fits of the BhCas4 structural model in the assigned Cas4 density, as outlined in the response below.

2) The Figures 2 and Figure 2—figure supplement 4 were not clear and the reviewers could not get a good handle on the orientation of the Cas4 proteins relative to the DNA. Figure 2 shows 20 Å EM reconstruction of the Cas1-2 and Cas1-2-4 complexes. At this resolution, the Cas1-2 complex model is probably OK and matches known structures. The extra "blob" in the Cas1-2-4 complexes is assigned to Cas4, however it is smaller than the smallest known Cas4, the monomeric protein from P. calidifontis. It also does not fit with the EM structure of the Cas1-4 complex previously reported by the same authors (Lee, 2018). Although the FeS cluster appeared to help, this assignment should be treated with caution. In particular, the orientation chosen for the modelled Cas4 looks close to arbitrary and should not be used as a basis for further interpretation about active site positioning.

As mentioned above, we agree with the reviewers that it is important to ensure the reader understands the ambiguity of our structural model. We had attempted to emphasize this ambiguity in the previous version of the manuscript, but the reviewers’ comments indicate that this could be improved further. We have now performed new structural modelling experiments that better emphasize this ambiguity. We have made several changes to Figure 2 and the figure supplements in an attempt to make the structural model clearer.

First, we would like to note that our data very strongly suggests that the extra density observed in the Cas4-Cas1-Cas2 complex is due to the presence of Cas4. For example, in Figure 1B-D, we show that Cas4 co-elutes with Cas1, Cas2 and DNA, while in Figure 1E-F, we demonstrate clearly visible extra density in 2D class averages that is only present when Cas4 is present in the sample. We therefore feel confident in our conclusion that this density can be assigned to Cas4.

The reviewers are correct that some of the PcCas4 structure lay outside the density in our previous version of Figure 2. There are multiple possible reasons for why the assigned Cas4 density is smaller than the PcCas4 structure. As noted above, the two structures are likely different as BhCas4 lacks the long N-terminal domain (~50 amino acids) and instead contains a short C-terminal domain (~20 amino acids). Although BhCas4 is overall longer, many of the insertion sequences are predicted to be in loop regions that are likely flexible and not resolved in the EM map (Figure 2 —figure supplement 4). Overall, these differences may result in a more compact BhCas4 structure in comparison to the PcCas4 we used in our previous modeling experiments. In addition, as we previously noted in the text, the Cas4 density segmentation may be incomplete due to low resolution, resulting in the apparently smaller Cas4 density. This effect of low resolution is reported in the publication describing the Segger tool used for our segmentation (Pintilie et al., 2010). It is therefore possible that Cas4 lies at least partially in the “Cas1” density.

This is now stated more explicitly in the text:

“The segmented density for Cas4 is smaller than the structure and parts of the structure fit into the Cas1 density, suggesting that the segmentation between Cas1 and Cas4 densities was incomplete due to low resolution as has been observed previously (Pintilie et al., 2010).”

We agree with the reviewers that the previous model based on the Fe_2_S_2_ cluster should be treated with caution. To ensure that this model did not lead to overinterpretation, we have removed it from the revised manuscript. We have instead included a new set of modelling experiments that better reflect the possible orientations Cas4 may adopt within the assigned density. We made two changes to our prior modeling experiments. First, we now use crystal structures (BhCas2 PDB 4ES3) or structural models (BhCas1 and BhCas4 predicted by Phyre2 server) of the actual *B. halodurans* sequences. As noted above, these predicted structural models may better reflect the size and shape of the actual sequences present in the complex. Second, we now use the “Fit to Segments” tool within the Segger package in UCSF Chimera to perform structural fitting (rather than the “Fit in Map” tool). This tool enables output of multiple possible top fits, allowing us to show several possible orientations for Cas4 to be docked into the assigned density. With our new modeling experiments, much of the BhCas4 structural model fits within the Cas4 density with reasonable cross-correlation scores (Figure 2—figure supplement 4). In all orientations, Cas4 is located in close proximity to the Cas1 active site, although the distance between the active sites varies. The new modeling and figures will provide a more comprehensive view of how Cas4 may interact with Cas1, and we hope this will address the reviewers’ concerns about overinterpretation of our structural data. The new modeling is shown in Figure 2 and Figure 2—figure supplement 4 and described in the Results section and Materials and methods section. Descriptions of how Cas1-Cas2 crystal structures fit into the Cas1-Cas2 density are now shown in Figure 2—figure supplement 3.

3) The absence of density corresponding to the CRISPR DNA is troublesome and needs to be discussed by the authors. What is the rationale of using the CRISPR leader-repeat-spacer instead of the prespacer duplex to promote the formation of the Cas4-Cas1-Cas2 complex? It appears that initially the authors reasoned that target-binding may promote complex formation, but their later data seems to argue that the Cas4-Cas1-Cas2 complex contacted the DNA rather nonspecifically, and through the prespacer-binding interface. If this is the case, to avoid possible misconceptions, the authors should make it clear that the contact does not reflect how Cas1-Cas2 recognizes the target sequence.

We agree that more discussion on the absence of DNA density is important and have added additional information on this in the text in paragraph six of subsection “Architecture of the Cas4-Cas1-Cas2 complex”. First, we note that it is well established that it is difficult to visualize nucleic acids by negative stain (e.g. Nogales et al., 2015 and Hochstrasser et al., 2014), so it is not unexpected that DNA is not visible in most of our EM densities. Our biochemical results show that the DNA co-elutes with the protein components of the complex (Figures 1C-D), which strongly suggests that the DNA is part of the complex, even though the density is not visible. The highest resolution Cas4-Cas1-Cas2 EM density (symmetrical complex with CRISPR hairpin target bound) does appear to have additional density along the prespacer binding surface, although we cannot unambiguously assign this density to the target DNA. Although we were not expecting the target DNA to bind along this surface, in hindsight, it seems likely that this is the preferred DNA-binding site for Cas1-Cas2. Interactions with the DNA backbone are non-specific along this surface in all known Cas1-Cas2-prespacer structures (Wang et al., 2015, Nuñez et al., 2015b, Xiao et al., 2017). In contrast, specific interactions with CRISPR DNA may be transient, and only stabilized upon formation of the half-site integration intermediate with the prespacer substrate. Consistently, there are no known structures of Cas1-Cas2 bound to only a CRISPR DNA, while structures of half-site and full-site intermediate-bound Cas1-Cas2 are available (Xiao et al., 2017, and PMID 28729350).

We have included discussion of this possibility in the Results section:

“Although we hypothesized that the CRISPR hairpin target would stabilize the Cas1-Cas2 complex by binding along the repeat-binding surface, our structural studies of complexes formed in the presence of the hairpin target suggest this is not the case. Notably, the prespacer-binding surface of Cas2 interacts with DNA through non-specific backbone contacts (Nuñez et al., 2015b; Wang et al., 2015; Xiao et al., 2017), and no structure of Cas1-Cas2 bound to CRISPR DNA alone has been previously reported. This could suggest that any dsDNA, including the CRISPR target, may preferentially bind along the prespacer-binding surface and that other DNA substrates binding in that location could promote stable complex formation.”

The reviewers are correct in interpreting our reasoning for using the hairpin target DNA. It has been challenging to isolate a Cas4-Cas1-Cas2 complex, due to the difficulty of working with Cas4, which is prone to aggregation and precipitation. We previously made extensive attempts to form the complex in the presence of prespacer and other types of dsDNA, and our attempts to do so with the CRISPR hairpin target proved the most successful. However, with recent improvements in our Cas4 purification and complex formation protocol, we have now been able to form the Cas4-Cas1-Cas2 complex in the presence of prespacer, rather than target DNA. These new experiments, including structures of symmetrical and asymmetrical Cas4-Cas1-Cas2prespacer are added to the manuscript in a new section titled “Formation of the Cas4-Cas1Cas2 complex with prespacer substrate” (subsection “Formation of the Cas4-Cas1-Cas2 complex with prespacer substrate”, Figure 1—figure supplement 2 and Figure 2—figure supplements 1, 2 and 6).

Major points on the biochemistry work:1) The biochemistry experiments in this manuscript were done using prespacer substrates containing a 'perfect' duplex with ss-overhangs provided either in cis or in trans. Such substrates are rarely found in vivo. In cases when the PAM is buried in the dsDNA region, can Cas4 process the dsDNA and correctly cleave after the 3'-PAM? Can the product be integrated into the CRISPR substrate?

In our previous publication, we tested Cas4 cleavage of several types of substrates, including dsDNA containing a PAM (Figure S2 in Lee et al. Mol Cell, 2018). We do not observe cleavage activity with any substrates except duplexes with 3ʹ overhangs. The new activity described in the current manuscript reinforces the idea that Cas4 only cleaves ssDNA, and indeed this is expected given the nature of the RecB nuclease domain. Thus, in order to cleave a PAM within a dsDNA, Cas4 would require a DNA unwinding activity that we have been unable to detect. These data indicate that additional factors or alternative conditions may be necessary to expose single-stranded DNA for Cas4 cleavage.

We have added the following sentences to the Discussion section to discuss this idea:

“Therefore, a duplex DNA containing single-strand overhangs is likely the optimal Cas4-Cas1-Cas2 substrate to enable efficient cleavage and substrate handoff. It remains unclear how these types of prespacers are formed. In other CRISPR-Cas sub-types, Cas4 has been demonstrated to have exonucleolytic activity that can lead to dsDNA unwinding (Lemak et al., 2013, 2014a; Zhang et al., 2012), although we have not observed robust exonuclease or dsDNA cleavage activity for the type I-C BhCas4 (Lee et al., 2018). It is possible that other host factors or conditions are required to unwind ends of dsDNA, or to expose single-stranded overhangs for precise cleavage by Cas4 in the type I-C system.”

2) What would happen when a ss-PAM is 1-2 nucleotide away (too far or too close) from the 5-nt spacing? Does the Cas4 cleavage dictate where the integration happens?

The question of PAM location within single-stranded overhang is interesting, and one we had not previously explored. We addressed the question of the effect of PAM location on single-strand overhang processing with a new set of experiments, described in our response to the next concern. For the second question, we addressed where integration happens in our previous publication (Figure 5 in Lee et al. Mol Cell, 2018). In that publication, we performed highthroughput sequencing experiments of half-site intermediate products to show that integration at the leader-repeat junction is highly precise. That experiment showed that the processing position did not affect the location of integration. However, integration at the repeat-spacer junction is much less specific. It may be interesting to determine whether the PAM position in the overhang affects the location of integration at the repeat-spacer junction. However, due to the relative rarity of integration at the repeat-spacer junction, this experiment requires high-throughput sequencing experiments. These experiments are costly and time consuming and we believe they will have to remain the subject of future experiments.

3) Figure 3 presents some nice data showing that prespacers are cleaved by Cas4 and that this is stimulated by the presence of ds CRISPR DNA. The data on ssDNA substrates, which is discussed in depth, is likely not that relevant to the adaptation process and activity is much weaker for these substrates. In the figure legend, part C is not described. This figure does lack an important control – use of a duplex substrate with ss overhangs that lacks a PAM sequence to discriminate between a ruler mechanism and a sequence specific mechanism for cleavage by Cas4 of the relevant prespacer structures.

There are several points in this concern that we will address individually:

“In the figure legend, part C is not described.”

We thank the reviewers for pointing out a typo in the figure legend for Figure 3 (panels C and D were described as panels D and E in the previous version of the manuscript). This is now corrected.

“Figure 3 presents some nice data showing that prespacers are cleaved by Cas4 and that this is stimulated by the presence of ds CRISPR DNA. The data on ssDNA substrates, which is discussed in depth, is likely not that relevant to the adaptation process and activity is much weaker for these substrates.”

To clarify, we do not observe stimulation of duplex prespacer cleavage in the presence of CRISPR DNA, although this was what we intended to test with this experiment. As observed in Figure 3B, the amount of product is similar in the 9th and 11th lane of the gel, indicating that CRISPR DNA does not enhance duplex prespacer cleavage. We were instead surprised to find that a dsDNA induces cleavage of a trans ssDNA. Although it is unclear whether this activity is relevant to the adaptation process, we believe that it informs on the mechanism of Cas4-Cas1Cas2, which is why it is emphasized in this figure and throughout the text. The experiments in Figure 3 establish that once the Cas1-Cas2 complex is stabilized by binding to a dsDNA, it can activate Cas4 for cleavage of ssDNA. This cleavage is indeed more efficient when the ssDNA is covalently attached to the Cas1-Cas2-bound duplex, which likely localizes the ssDNA in close proximity to the Cas4 active site. However, the fact that Cas4 can cleave a trans ssDNA demonstrates that the single-stranded substrate is not positioned in the Cas4 active site solely based on the interaction with Cas1-Cas2. Instead, Cas4 can bind and cleave any ssDNA once activated by Cas1-Cas2. We have added more clarification on this point throughout the text.

We also added the following to the Discussion to emphasize the implications of the ssDNA cleavage:

“Surprisingly, Cas4 processing can occur at several locations within both trans and cis ssDNA, as long as enough space is available between cleavage sites and the end of a duplex. Importantly, we do not observe Cas4-dependent PAM processing in the absence of Cas1Cas2. These observations suggest that the Cas4 active site is activated for cleavage based on interactions with Cas1-Cas2. Once activated, Cas4 can sample any region of a single-stranded DNA that is accessible to the active site. This feature could allow more flexibility in defining prespacer substrates due to the ability of Cas4 to search for PAM sequences within single-stranded overhangs.”

“This figure does lack an important control – use of a duplex substrate with ss overhangs that lacks a PAM sequence to discriminate between a ruler mechanism and a sequence specific mechanism for cleavage by Cas4 of the relevant prespacer structures.”

We performed the control experiment suggested by the reviewers in a previous publication (Figure 3 in Lee et al. Mol Cell, 2018,) and did not observe cleavage when no PAM is present in the single-stranded overhang. We now explicitly mention this in the Results section:

“Importantly, we previously showed that cleavage of the duplex prespacer is dependent on having a PAM present in the single-stranded overhang, indicating that cleavage occurs in a PAM-dependent manner (Lee et al., 2018).”

However, the question of a potential ruler mechanism is important and one we had not entirely ruled out in our previous experiments. To address this question, as well as the first question proposed in concern #2, we have performed additional processing experiments using a panel of duplex substrates in which the PAM position is varied within the single-stranded overhang. Consistent with our other data, we observe that the PAM position does determine where processing occurs, indicating a PAM-dependent cleavage mechanism rather than a ruler mechanism. However, the position of the PAM is important for cleavage, as a PAM sequence that is too close to the duplex was not processed efficiently. This suggests that the single-stranded overhang must be long enough to stretch from the end of the duplex-binding region of Cas1-Cas2 to the active site of Cas4, but longer single-stranded regions can be cleaved at any PAM-containing region. This is reminiscent of the trans-ssDNA cleavage observed in our other experiments, and suggests that the Cas1-Cas2 complex does not position the ssDNA in the Cas4 overhang as long as the PAM is far enough away from the duplex. These experiments are described in a new Results section “Cas4 cleavage depends on PAM location within single-stranded overhangs” (Figure 6).

4) Figure 4 and 5 show convincing data supporting PAM-dependent cleavage activity by Cas1-2-4 on ssDNA substrates. This provides important new information on PAM removal by Cas4 during the capture process. It is unclear why this work has not been done with the relevant substrate, the gapped duplex shown in Figure 3B, which is cleaved more efficiently that non-relevant ssDNA. It would considerably strengthen the paper if the authors could provide some evidence that the same PAM-specific cleavage is observed with relevant duplex substrates, and this should not be hard to do.

We had multiple reasons for using the ssDNA in our sequencing gel experiments. First, using long single-stranded substrates enabled the introduction of primer-binding sites for Sanger sequencing, allowing us to run the sequencing lanes alongside processing reactions for unambiguous detection of the cleavage site. Second, we wanted to demonstrate that the only sites cleaved by Cas4 are PAM sites by using a panel of different sequences. Finally, we wanted to test whether multiple PAM sites could be cleaved within a single substrate. These latter two experiments were more easily done with the ssDNA, where the whole DNA would be subject to potential cleavage by Cas4. In contrast, only the single-strand overhang is subject to cleavage by Cas4 in the duplex substrate, limiting the amount of sequence we could interrogate for cleavage. Therefore, by using this experimental system, we were able to observe Cas4 specificity more effectively. We added the following sentence to better explain our reasoning for this line of experimentation:

“By using ssDNA substrates, we could more readily test sequences containing multiple PAM sites or a variety of sequences, and additionally include primer-binding sites for Sanger sequencing reactions.”

Unfortunately, the experiment suggested by the reviewers is not as straightforward as they propose given our recent personnel changes, and we believe we have already demonstrated that the single-strand overhangs are cleaved at the same position as the ssDNA. In Figure 3B, we show that the single-stranded and duplex substrates are cleaved at identical positions, indicating that the duplex is also cleaved in a PAM-specific manner. In addition, we have now performed the experiments described above in which cleavage site within single-strand overhang varies based on the location of the PAM. Overall, these experiments show that Cas4 only cleaves single-stranded sequences in a PAM-dependent manner, and that these substrates are cleaved in the same position regardless of whether they are fully single-stranded or partially duplexed.